# Chronic optogenetic induction of stress granules is cytotoxic and reveals the evolution of ALS-FTD pathology

**Peipei Zhang[1], Baochang Fan[1], Peiguo Yang[1], Jamshid Temirov[1], James Messing[2], Hong Joo Kim[1], J Paul Taylor[2]***

[1]Department of Cell and Molecular Biology, St. Jude Children's Research Hospital, Memphis, United States; [2]Howard Hughes Medical Institute, Chevy Chase, United States

**Abstract** Stress granules (SGs) are non-membrane-bound RNA-protein granules that assemble through phase separation in response to cellular stress. Disturbances in SG dynamics have been implicated as a primary driver of neurodegenerative diseases, including amyotrophic lateral sclerosis (ALS) and frontotemporal dementia (FTD), suggesting the hypothesis that these diseases reflect an underlying disturbance in the dynamics and material properties of SGs. However, this concept has remained largely untestable in available models of SG assembly, which require the confounding variable of exogenous stressors. Here we introduce a light-inducible SG system, termed OptoGranules, based on optogenetic multimerization of G3BP1, which is an essential scaffold protein for SG assembly. In this system, which permits experimental control of SGs in living cells in the absence of exogenous stressors, we demonstrate that persistent or repetitive assembly of SGs is cytotoxic and is accompanied by the evolution of SGs to cytoplasmic inclusions that recapitulate the pathology of ALS-FTD.

**Editorial note:** This article has been through an editorial process in which the authors decide how to respond to the issues raised during peer review. The Reviewing Editor's assessment is that all the issues have been addressed (see decision letter).

DOI: https://doi.org/10.7554/eLife.39578.001

*For correspondence:
jpaul.taylor@stjude.org

## Introduction

Genetic, pathologic, biophysical, and cell biological evidence has implicated disturbances in stress granules as a primary driver of several common neurodegenerative diseases, including ALS, FTD, and inclusion body myopathy (IBM) (*Molliex et al., 2015*; *Mackenzie et al., 2017*; *Taylor et al., 2016*; *Lee et al., 2016*; *Ramaswami et al., 2013*; *Buchan et al., 2013*; *Patel et al., 2015*; *Hackman et al., 2013*). These diseases show substantial clinical and genetic overlap and share the hallmark histopathological feature of cytoplasmic inclusions composed of RNA-binding proteins and other constituents of ribonucleoprotein (RNP) granules in affected neurons and muscle cells. A prominent feature of this end-stage cytoplasmic pathology is ubiquitinated and phosphorylated forms of TDP-43, although a host of other proteins co-localize with these pathological inclusions, including related RNA-binding proteins and ubiquitin-binding proteins such as SQSTM1, UBQLN2, OPTN, and VCP (*Neumann et al., 2006*; *Mackenzie et al., 2007*; *Mackenzie and Neumann, 2016*; *Williams et al., 2012*; *Deng et al., 2011*).

Many mutations that cause ALS-FTD and/or IBM impact RNA-binding proteins that are building blocks of stress granules (e.g., TDP-43, hnRNPA1, hnRNPA2B1, hnRNPDL, TIA1, matrin 3, and FUS). Furthermore, these mutations largely cluster in low-complexity, intrinsically disordered regions (IDRs) and in many cases have been shown to change the dynamic properties of stress granules

(*Mackenzie et al., 2017*; *Hackman et al., 2013*; *Kim et al., 2013*; *Liu-Yesucevitz et al., 2010*). Another set of disease-causing mutations impact ubiquitin-binding proteins (e.g., UBQLN2, VCP, p62/SQSTM1, and OPTN) whose functions intersect with disassembly and/or clearance of stress granules (*Buchan et al., 2013*; *Dao et al., 2018*; *Chitiprolu et al., 2018*). Furthermore, pathological poly-dipeptides arising from repeat-expanded *C9orf72*, the most common genetic cause of ALS-FTD, insinuate into stress granules and other membrane-less organelles, perturbing their dynamics and/or functions (*Lee et al., 2016*; *Boeynaems et al., 2017*). Several ALS-, FTD-, and IBM-causing mutations cause aberrant phase separation and change the biophysical and material properties of stress granules, generally resulting in poorly dynamic membrane-less organelles that, it has been suggested, may evolve into the cytoplasmic pathology found in end-stage disease (*Mackenzie et al., 2017*; *Buchan et al., 2013*; *Kim et al., 2013*). However, no direct evidence has demonstrated that perturbation of phase separation is sufficient to drive neurotoxicity or that ALS-FTD-associated inclusions represent the endpoint of a formerly dynamic stress granule. Moreover, capitalizing on mechanistic links between stress granules and disease to identify therapeutic targets has been limited by models employing exogenous stressors (e.g., heat shock, arsenite) to initiate stress granule assembly, with numerous nonspecific and pleiotropic effects.

Stress granules are comparatively large (~50 nm to ~3 µm) biomolecular condensates that rapidly form in the cytoplasm in response to a wide variety of stressors (*Protter and Parker, 2016*; *Panas et al., 2016*). Like other RNP granules, stress granules are believed to arise at least in part through liquid-liquid phase separation (LLPS), a biophysical phenomenon in which RNA-protein complexes separate from the surrounding aqueous cytoplasm to create a functional cellular compartment with liquid properties (*Molliex et al., 2015*; *Protter and Parker, 2016*). Stress granule assembly is a complex process that involves a cascade of events, including the dismantling of polysomes and reorganization of mRNPs into discrete cytoplasmic foci that contain >400 different protein constituents (*Jain et al., 2016*; *Markmiller et al., 2018*; *Youn et al., 2018*) and >1800 different RNAs (*Khong et al., 2017*). The assembly of RNP granules, including stress granules, is driven in part by the collective behavior of many macromolecular interactions, including RNA-RNA interactions, protein-RNA interactions, conventional interactions between folded protein domains, as well as weak, transient interactions mediated by low complexity IDRs of proteins – particularly those present in RNA-binding proteins (*Banani et al., 2017*). While there is consensus about the major underlying forces that drive RNP granule assembly, the precise mechanisms that orchestrate the assembly of distinct types of RNP granules are largely unknown, although general principles have been suggested by in vitro studies (*Banani et al., 2016*). In this conceptual framework, RNP granules and other biomolecular condensates are established and maintained by a small number of essential constituents defined as *scaffolds*, whereas the remaining constituents are considered *clients* (*Banani et al., 2016*).

Although at least six proteins have been suggested to be 'essential' elements of stress granules (*Markmiller et al., 2018*; *Youn et al., 2018*; *Kedersha et al., 2016*; *Gilks et al., 2004*; *Kwon et al., 2007*), until recently it was unknown which of these proteins (if any) are true scaffolds for stress granules. In related work that informs the study presented here, we performed a whole-genome screen that identified G3BP as a uniquely essential scaffold in stress granule assembly (Yang, Mathieu *et al.*, unpublished). Moreover, we found that an oligomerization domain within G3BP that is essential to stress granule assembly could be functionally replaced by heterologous oligomerization domains, which suggested the possibility of engineering temporal and spatial control of stress granule assembly without the confounding influences of stress (Yang, Mathieu *et al.*, unpublished). We built upon a previously described system, termed 'OptoDroplets,' which uses optogenetic oligomerization of proteins as a means to control intracellular LLPS (*Shin et al., 2017*). In this system, light-sensitive chimeric proteins are assembled from the IDRs of various RNP granule proteins combined with the light-sensitive oligomerization domain of *Arabidopsis thaliana* cryptochrome 2 (CRY2) photolyase homology region (PHR) to generate fusion proteins that undergo LLPS in living cells upon blue light activation. Whereas enforced aggregation of IDRs drives LLPS and thereby leads to OptoDroplet formation, it is not anticipated that droplets formed by the IDRs of any given RNP granule protein will initiate the full cascade of *bona fide* RNP granule assembly. However, we reasoned that adapting this OptoDroplet system might provide a means of testing the hypothesis that enforced LLPS of key stress granule constituents could distinguish between stress granule scaffolds and clients, in which LLPS of a *scaffold* protein would initiate a process that faithfully reconstitutes the assembly of a

stress granule, whereas LLPS of a *client* protein would not. Moreover, if we succeeded in optical induction of stress granules, it would provide the first opportunity to examine the consequences of protracted stress granule assembly without the confounding variable of exogenous stress.

Herein we report that light-based activation of Opto-G3BP1, a chimeric protein assembled from the IDR and RNA-binding domain of G3BP1 combined with CRY2$_{PHR}$, initiated the rapid assembly of dynamic, cytoplasmic, liquid granules that were composed of canonical stress granule components, including PABP, TDP-43, TIA1, TIAR, eIF4G, eIF3η, ataxin 2, GLE1, FUS, and polyadenylated RNA, thereby establishing the identity of G3BP1 as a scaffold protein for stress granules. To differentiate these complex assemblies formed by LLPS of the scaffold protein G3BP1 from the relatively homogenous clusters formed by LLPS of client proteins, we termed these structures OptoGranules. Importantly, we found that persistent or repetitive assembly of OptoGranules is cytotoxic and is accompanied by the evolution of these granules to neuronal cytoplasmic inclusions characteristic of ALS-FTD.

## Results

To test whether optogenetically induced LLPS of a stress granule scaffold protein could faithfully reconstitute the assembly of a *bona fide* stress granule, we first investigated G3BP1 as a potential scaffold protein. G3BP1 (and its close paralog G3BP2) has been suggested to be an essential nucleator of stress granule assembly (*Kedersha et al., 2016*), and a genome-wide screen recently identified G3BP1/2 as a uniquely essential protein for stress granule assembly (Yang, Mathieu *et al.*, unpublished). G3BP1 has an N-terminal 142-amino acid dimerization domain, termed the NTF2L domain, that is essential for nucleation of stress granule assembly. Remarkably, the NTF2L domain can be replaced by generic dimerization domains, and the resulting chimeric proteins are able to fully nucleate stress granule assembly in living cells (Yang, Mathieu *et al.*, unpublished). Thus, the domain architecture of G3BP1 is ideal for engineering light-inducible stress granule assembly by replacing the NTF2L domain of G3BP1 with the blue light-dependent dimerization domain CRY2$_{PHR}$ in frame with the fluorescent proteins mCherry or mRuby. We named this construct 'Opto-G3BP1' and also created an 'Opto-Control' construct referring to CRY2$_{PHR}$-mCherry (or mRuby) alone (*Figure 1a*).

We next generated U2OS cell lines stably expressing Opto-Control or Opto-G3BP1 constructs with comparable expression levels of Opto-G3BP1 and endogenous G3BP1 (*Figure 1—figure supplement 1a*). Within seconds of blue light activation, Opto-G3BP1 in U2OS cells assembled into numerous, spherical cytoplasmic granules that exhibited liquid behaviors (*Figure 1b* and *Videos 1* and *2*). A 5-millisecond pulse of blue light using a 488 nm vector laser (~2.5 MW/cm$^2$) was sufficient to initiate robust induction of these cytoplasmic granules, and these granules spontaneously disassembled over a period of approximately 5 min (*Figure 1b,c*). These granules were highly dynamic, exhibiting liquid behaviors such as fusion to form larger granules and relaxation to a spherical shape (*Video 2*). In contrast, under the same conditions, Opto-Control expression remained diffuse, with a modest amount of nuclear and cytoplasmic clusters (*Figure 1b* and *Video 1*). To confirm the dynamic nature of the optically induced granules, we performed fluorescence recovery after photobleaching (FRAP) to monitor recovery rates and mobile fractions of individual granules (*Figure 1d–f*), finding that these properties were very similar between Opto-G3BP1 and the conventional stress granule marker G3BP1-GFP. Furthermore, Opto-G3BP1 localized to spontaneous stress granules induced by expression of ALS mutant proteins (FUS R521C, TDP-43 ΔNLS, TIA1 A381T) even in the absence of blue light activation, demonstrating that the Opto-G3BP1 protein behaves similarly to endogenous G3BP1 (*Figure 1—figure supplement 1b*).

To further define the relationship between light-induced Opto-G3BP1 granules and stress-induced stress granules, we next examined their composition. Employing live cell imaging, we documented the dynamic recruitment of the stress granule marker GFP-TIA1 into optically induced granules following light-induced assembly (*Figure 1g* and *Video 3*). In contrast, clusters of Opto-Control (olig), a modified form of the Opto-Control construct designed to produce abundant aggregates, did not recruit GFP-TIA1 (*Figure 1—figure supplement 1c*), nor did they show dynamic behavior by FRAP (*Figure 1—figure supplement 1d–f*).

We further examined the composition of optically induced Opto-G3BP1 granules by staining activated cells for additional stress granule components. In these experiments, we employed a blue-light

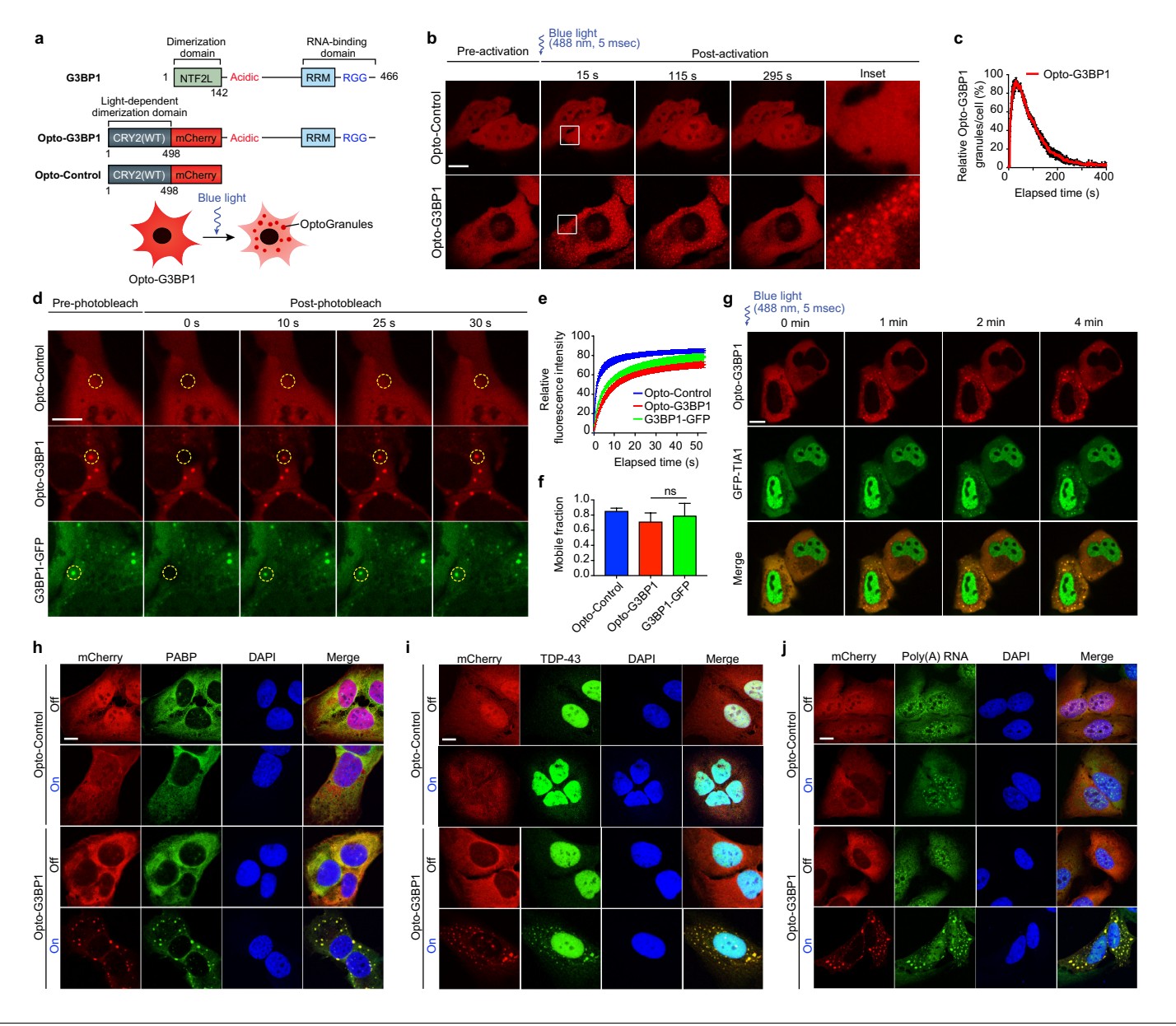

**Figure 1.** OptoGranules are light-inducible dynamic stress granules. (a) Design of Opto-G3BP1 and Opto-Control constructs. (b) U2OS cells stably expressing Opto-Control or Opto-G3BP1 were stimulated with a single 5-msec pulse of 488 nm blue light (power density ~2.5 MW/cm$^2$) in a defined ROI. Representative images are shown from n = 3 independent experiments. (c) Quantification of data in cells treated as in (b). Five cells with similar expression levels were counted. Granule numbers are shown relative to the granule number at the peak of OptoGranule assembly. Error bars represent s.e.m. (d-f) U2OS cells were stably transfected with Opto-Control or Opto-G3BP1, or stable Opto-G3BP1 cells were transiently transfected with G3BP1-GFP, and stimulated with a blue-light laser (power density ~4.5 W/cm$^2$) for 3 mins. Regions marked with yellow circles were photobleached and monitored for fluorescence recovery. Data are shown as representative images (d), relative fluorescence intensity of photobleached region over time (e), and relative mobile fraction derived from (e) (f). For (e, f) n = 15 cells for Opto-Control; n = 12 for Opto-G3BP1; n = 14 for G3BP1-GFP. Data are representative of n = 3 independent experiments. Data shown as mean + s.d. ns, not significant by one-way ANOVA with Dunnett's test. (g) U2OS cells transiently transfected with Opto-G3BP1 and the stress granule marker GFP-TIA1 were stimulated with a blue-light laser (power density ~2.5 MW/cm$^2$) for 5 msec. Cells were sequentially imaged by 561 nm and 488 nm channels; we note that the 488 nm channel used for imaging also activates Opto-G3BP1 (power density 2.2 W/cm$^2$). Representative images are shown from n = 3 independent experiments. (h-j) U2OS cells stably expressing Opto-Control or Opto-G3BP1 constructs were stimulated for 6 hr without or with continuous blue light (~2 mW/cm$^2$) using custom-made LED arrays for global activation. Cells were immunostained with PABP antibody (h), TDP-43 antibody (i), or RNA fluorescence in situ hybridization using FAM-labelled oligo (dT)20 as a probe (j). Scale bars, 10 μm in all micrographs.

DOI: https://doi.org/10.7554/eLife.39578.002

*Figure 1 continued*

The following figure supplements are available for figure 1:

**Figure supplement 1.** Opto-G3BP1 assembles light-dependent cytoplasmic clusters.
DOI: https://doi.org/10.7554/eLife.39578.003
**Figure supplement 2.** OptoGranules are light-inducible stress granules.
DOI: https://doi.org/10.7554/eLife.39578.004
**Figure supplement 3.** OptoGranules rescue stress granule formation in *G3BP1/2* double KO cells.
DOI: https://doi.org/10.7554/eLife.39578.005
**Figure supplement 4.** OptoDroplets are not stress granules.
DOI: https://doi.org/10.7554/eLife.39578.006

LED array that permitted global activation of a larger number of cells. This LED array has a much lower energy density (~2 mW/cm$^2$) than the laser used for dynamic imaging, drives less oligomerization of CRY2$_{PHR}$, and therefore offers slower kinetics to facilitate monitoring of the recruitment of granule components over time. In cells expressing Opto-G3BP1, but not Opto-Control, all stress granule components that we examined, including PABP, TDP-43, TIA1, TIAR, eIF4G, eIF3η, ataxin 2, GLE1, and FUS, were recruited to optically induced granules (*Figure 1h–i* and *Figure 1—figure supplement 2a–g*). Since stress granules represent assemblies of mRNA as well as protein (*Panas et al., 2016*; *Kedersha et al., 1999*), we used fluorescent in situ hybridization (FISH) with fluorescently conjugated oligo(dT) probes to examine whether polyadenylated mRNAs were present in these optically induced granules as in canonical stress granules. We found that polyadenylated mRNAs were recruited into optically induced granules that assembled after blue light stimulation but showed no relocalization in cells expressing Opto-Control (*Figure 1j*). These findings indicate that optically induced Opto-G3BP1 granules are stress granules composed of mRNAs and RNA-binding proteins, including ALS-associated proteins such as TDP-43, ataxin 2, GLE1, FUS, and TIA1.

Consistent with prior reports, knockout of endogenous *G3BP1* and *G3BP2* in U2OS cells abolished stress granule assembly in response to arsenite (*Kedersha et al., 2016*) (*Figure 1—figure supplement 3a*). When introduced into these *G3BP1/G3BP2* double knockout cells, Opto-G3BP1 (or the analogous chimeric protein Opto-G3BP2) substantially restored stress granule assembly in response to blue light activation, demonstrating that the scaffolding activity of G3BP1 in the chimeric protein is functionally intact (*Figure 1—figure supplement 3b,c*).

The initiation of stress granule assembly in response to enforced LLPS of G3BP1 differs from prior observations made regarding Opto-Droplets, which do not typically represent assembly of complex, physiologically assembled membrane-less organelles (*Shin et al., 2017*). To examine this in more detail, we generated a series of optically inducible chimeric proteins by generating constructs in which CRY2$_{PHR}$-mCherry was fused with stress granule constituent proteins, including full-length or truncated versions of FUS, TDP-43, and TIA1. Opto-FUS [CRY2$_{PHR}$-mCherry-FUS(IDR)] and Opto-TDP-43 [CRY2$_{PHR}$-mCherry-TDP-43(IDR)] did assemble into droplets with blue light activation, as previously reported (*Shin et al., 2017*), but these Opto-Droplets did not recruit stress granule constituents commonly used as markers, including G3BP1 and PABP (*Figure 1—figure supplement 4a–c*), and were also negative for stress granule constituents VCP, SQSTM1, and the related protein OPTN (*Figure 1—figure supplement 4d,e*).

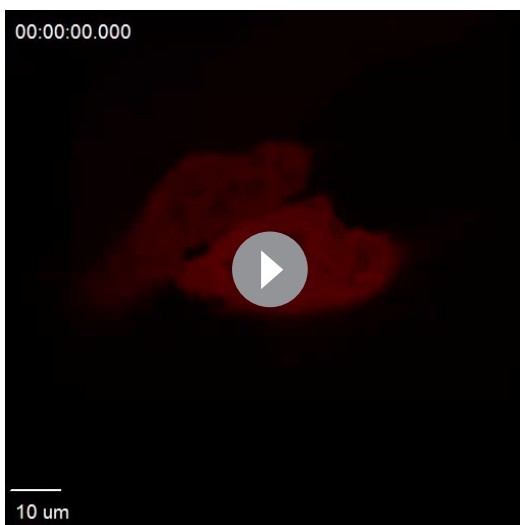

**Video 1.** Opto-Control fails to assemble light-dependent cytoplasmic clusters. U2OS cells stably expressing Opto-Control were stimulated with a single 5-msec pulse of 488 nm blue light (power density ~2.5 MW/cm$^2$) in a defined ROI. See *Video 2* for corresponding Opto-G3BP1 condition.
DOI: https://doi.org/10.7554/eLife.39578.007

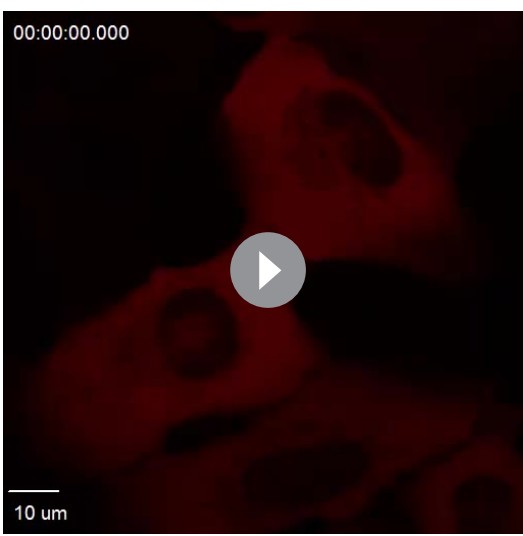

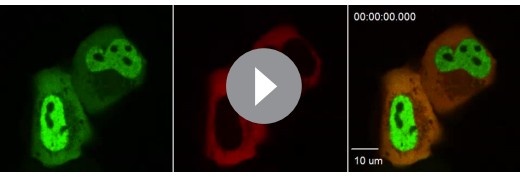

**Video 3.** Dynamic recruitment of the stress granule marker GFP-TIA1 into light-induced Opto-G3BP1 granules. U2OS cells transiently transfected with Opto-G3BP1 and the stress granule marker GFP-TIA1 were stimulated with a blue-light laser (power density ~2.5 MW/cm$^2$) for 5 msec. Cells were sequentially imaged by 561 nm and 488 nm channels; we note that the 488 nm channel used for imaging (power density 2.2 W/cm$^2$) also activates Opto-G3BP1.
DOI: https://doi.org/10.7554/eLife.39578.009

**Video 2.** Opto-G3BP1 assembles light-dependent cytoplasmic clusters. U2OS cells stably expressing Opto-G3BP1 were stimulated with a single 5-msec pulse of 488 nm blue light (power density ~2.5 MW/cm$^2$) in a defined ROI. Opto-G3BP1 assembles highly dynamic, liquid-like cytoplasmic granules, and these granules spontaneously disassemble over a period of approximately 5 min. See **Video 1** for corresponding Opto-Control condition.
DOI: https://doi.org/10.7554/eLife.39578.008

Similarly, constructs containing the IDR and RNA recognition motifs of FUS or TDP-43 [CRY2$_{PHR}$-mCherry-FUS (1–371 aa); CRY2$_{PHR}$-mCherry-TDP-43 (106–414 aa)] assembled into droplets upon blue light activation, but these droplets were also negative for stress granule markers (*Figure 1—figure supplement 4f,g*). Expression of Opto-constructs using full-length FUS or TDP-43 [CRY2$_{PHR}$-mCherry-FUS(FL); CRY2$_{PHR}$-mCherry-TDP-43(FL)] did not produce stress granules with blue light activation (*Figure 1—figure supplement 4f,g*). Finally, Opto-TIA1, which represents fusion of CRY2 with TIA1 (CRY2$_{PHR}$-mCherry-TIA1), also assembled into droplets with blue light activation, but did not drive the assembly of stress granules, as illustrated by lack of colocalization with stress granule markers (*Figure 1—figure supplement 4h–j*). These data indicate that RNP granule assembly cannot be driven by enforced LLPS of any random constituent, but depends upon specific constituents. This conclusion is consistent with the proposition that LLPS initiated by scaffold proteins (e.g., G3BP1) has the capacity to initiate a membrane-less organelle, whereas client proteins (e.g., FUS, TDP-43, TIA1), even when forced to undergo LLPS, cannot reconstitute such a complex assembly (*Banani et al., 2017*). Thus, we termed Opto-G3BP1-induced stress granules 'OptoGranules' to distinguish them from OptoDroplets.

Phase transitions are highly dependent on protein concentration, and we therefore hypothesized that the induction of OptoGranule assembly would be dependent on the local concentration of activated G3BP1, similar to the concentration-dependent formation of light-activated OptoDroplets (*Shin et al., 2017*). To test this prediction, we controlled the local G3BP1 molecular concentration by modulating either the intensity of the activating blue light or the expression level of the Opto-G3BP1 construct. As predicted, we observed a strong positive correlation between blue light intensity and induction of OptoGranules (*Figure 2a,b*) and, independently, a strong positive correlation between Opto-G3BP1 expression level and induction of OptoGranules (*Figure 2c,d*). Thus, the OptoGranule system is highly tunable, a useful feature for a variety of studies.

We next examined the role of upstream events in OptoGranule formation and compared these to the cellular triggers associated with conventional stress granule assembly. Given that conventional stress granule formation is typically linked to the disassembly of translating polysomes (*Panas et al., 2016*), we tested whether polysome disassembly is required for OptoGranule formation. We determined that treatment with cycloheximide, which traps translating mRNAs within polysomes, strongly mitigated the formation of arsenite-induced stress granules and the formation of light-induced OptoGranules (*Figure 2e,f*), indicating that OptoGranule formation is dependent on polysome disassembly and further illustrating commonality with conventional stress granules. We next tested the

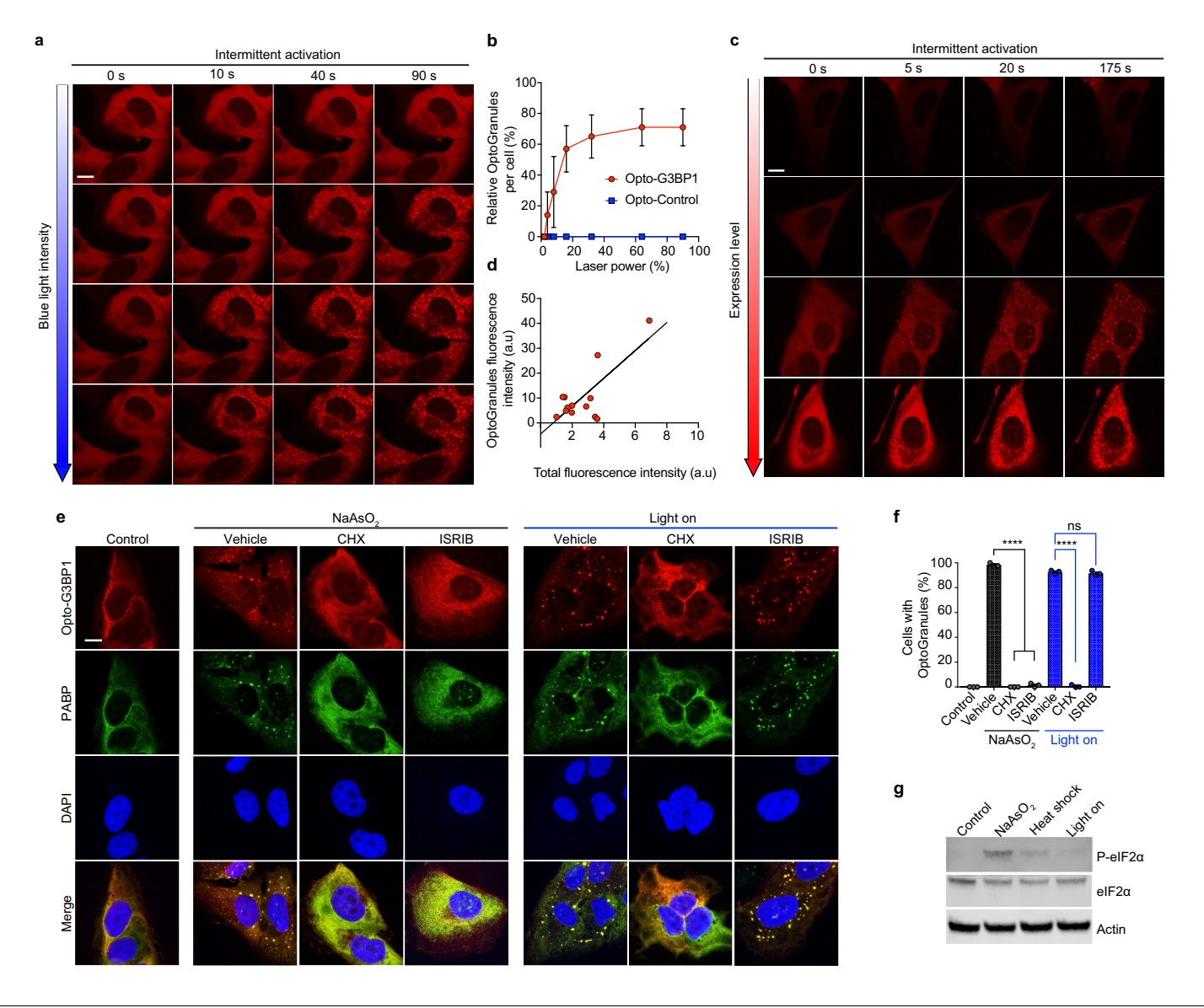

**Figure 2.** OptoGranule formation is dependent on the local concentration of activated G3BP1 and dependent on polysome disassembly, but independent of eIF2α phosphorylation. (**a**) U2OS cells stably expressing Opto-G3BP1 were intermittently exposed to a blue-light laser (488 nm) for activation followed by image acquisition with a 561 nm channel. Blue light intensity was sequentially increased from top to bottom (488 nm power density measurement from top to bottom: 1%, 0.02 W/cm$^2$; 5%, 0.04 W/cm$^2$; 25%, 0.95 W/cm$^2$; 75%, 5.5 W/cm$^2$). Representative images are shown from n = 3 independent experiments. (**b**) Quantification of data in cells treated as in (**a**). Error bars represent s.d. (**c**) U2OS cells with different expression levels of Opto-G3BP1 were intermittently exposed to a 488 nm blue-light laser (90% laser power, power density 6.3 W/cm$^2$) followed by image acquisition with a 561 nm channel. Relative expression levels from top to bottom: 0.19, 0.32, 0.78 and 1 a.u. Representative images are shown from n = 3 independent experiments. (**d**) Quantification of data in cells treated as in (**c**). (**e**) U2OS cells stably expressing Opto-G3BP1 were pre-treated with cycloheximide (CHX) or ISRIB for 30 min and then exposed to 45 min of sodium arsenite (0.5 mM NaAsO$_2$) or 6 hr of continuous blue light (~2 mW/cm$^2$) using custom-made LED arrays for global activation, and immunostained with PABP antibody. (**f**) Quantification of granule-positive cells from (**e**). Data are shown as mean ± s.e.m. from n = 3 independent experiments. ****p<0.0001; ns, not significant by one-way ANOVA with Tukey's post-test. (**g**) Immunoblot showing phosphorylated eIF2α (P-eIF2α), eIF2α, and actin levels in cells treated with sodium arsenite (0.5 mM NaAsO$_2$) for 45 min, exposed to 42°C heat shock for 1 hr, or activated with blue light for 6 hr. See also *Figure 2—figure supplement 1* for sequential probe images. Scale bars, 10 µm in all micrographs.

DOI: https://doi.org/10.7554/eLife.39578.010

The following figure supplement is available for figure 2:

**Figure supplement 1.** OptoGranule formation is independent of eIF2α phosphorylation.

*Figure 2 continued on next page*

*Figure 2 continued*

DOI: https://doi.org/10.7554/eLife.39578.011

role of eIF2α phosphorylation, which integrates stress granule formation downstream of a variety of stressors, such as arsenite and heat shock (*Panas et al., 2016*). We used the small molecule ISRIB, which binds eIF2B and interrupts eIF2α-mediated translational control (*Sidrauski et al., 2015*). We found that formation of arsenite-induced stress granules was blocked by ISRIB, as previously documented (*Sidrauski et al., 2015*), whereas the formation of light-induced OptoGranules was unaffected by ISRIB treatment (*Figure 2e,f*). Consistent with this finding, Western blotting also showed minimal phosphorylated eIF2α accompanying OptoGranule assembly (*Figure 2g*, *Figure 2—figure supplement 1*). Thus, OptoGranule formation depends upon the recruitment of mRNPs from polysomes, but this assembly occurs downstream and independent of the evolutionarily conserved integrated stress response regulated by eIF2α. This observation is consistent with the notion that OptoGranule formation is not driven by the classic signaling pathway for stress granule formation, which increases the concentration of free, uncoated RNA in the cytoplasm, but rather by oligomerization of G3BP1, which increases the valency for RNA binding.

Given the accumulating evidence that disturbance of membrane-less organelles such as stress granules may contribute to the initiation or progression of disease, we hypothesized that discrete disturbance in the dynamics or material properties of stress granules should be sufficient to cause cytotoxicity and recapitulate the pathognomonic features of specific diseases. To test this prediction, we examined the consequences of chronic OptoGranule assembly. First, we examined the consequences of continuous blue light activation in cells expressing Opto-G3BP1 or Opto-Control. We found that continuous induction of OptoGranules using a blue-light LED array resulted in progressive loss of cell viability reflected by progressive loss of crystal violet staining and depletion of ATP levels (*Figure 3a,b*). However, we also noted that chronic exposure to blue light resulted in a moderate amount of cytotoxicity in cells expressing Opto-Control or parental U2OS cells (*Figure 3b*). Although cells expressing Opto-G3BP1 exhibited significantly greater loss of viability upon exposure to blue light than cells expressing Opto-Control or parental U2OS cells, we sought to eliminate this potentially confounding background toxicity.

We therefore used live, confocal-based imaging to monitor cell viability in real time during 488 nm vector laser-induced OptoGranule induction. We first used a paradigm consisting of 2 s blue light pulses alternating with 12 s of rest, which drove robust OptoGranule assembly but left insufficient time for granules to disassemble prior to the next light pulse, resulting in persistent OptoGranule assembly (*Figure 3c*). Interestingly, persistent OptoGranule assembly under these conditions (2 s on, 12 s off) resulted in significant loss of viability in cells expressing Opto-G3BP1, with comparatively greatly reduced toxicity in cells expressing Opto-Control (*Figure 3c*). We next established a paradigm that further minimized blue light exposure, using a 10 s blue light pulse followed by 10 min of rest, which was sufficient to initiate robust assembly of OptoGranules that were able to fully disassemble prior to the next light pulse (*Figure 3d*). This paradigm of chronic, intermittent OptoGranule assembly, which may more closely reflect physiological, chronic, intermittent stress, greatly mitigated background toxicity due to blue light exposure and revealed significant toxicity in cells expressing Opto-G3BP1 compared to cells expressing Opto-Control (*Figure 3d*). The cell death associated with chronic intermittent OptoGranule assembly progressed more slowly than the cell death caused by chronic persistent OptoGranule assembly, although this difference did not reach statistical significance (*Figure 3—figure supplement 1a*). Taken together, these results demonstrate that chronic persistent or chronic intermittent stress granule assembly is intrinsically cytotoxic, independent of exogenous stressors.

Disease pathology in tissue from patients with ALS and FTD is marked by deposits of ubiquitin, ubiquitin-binding proteins, and TDP-43 that is cleaved and abnormally phosphorylated at Ser409/410 (*Neumann et al., 2009*). Newly formed OptoGranules were easily distinguished from the pathology present in late-stage ALS and FTD. Although OptoGranules were initially immunopositive for TDP-43 (as are conventional arsenite-induced stress granules) and ubiquitin, they were immunonegative for phospho-TDP-43 and ubiquitin-binding proteins (*Figure 3e–i*). OptoGranules were, however, immunopositive for staining by anti-A11, a conformation-specific antibody that recognizes amyloid

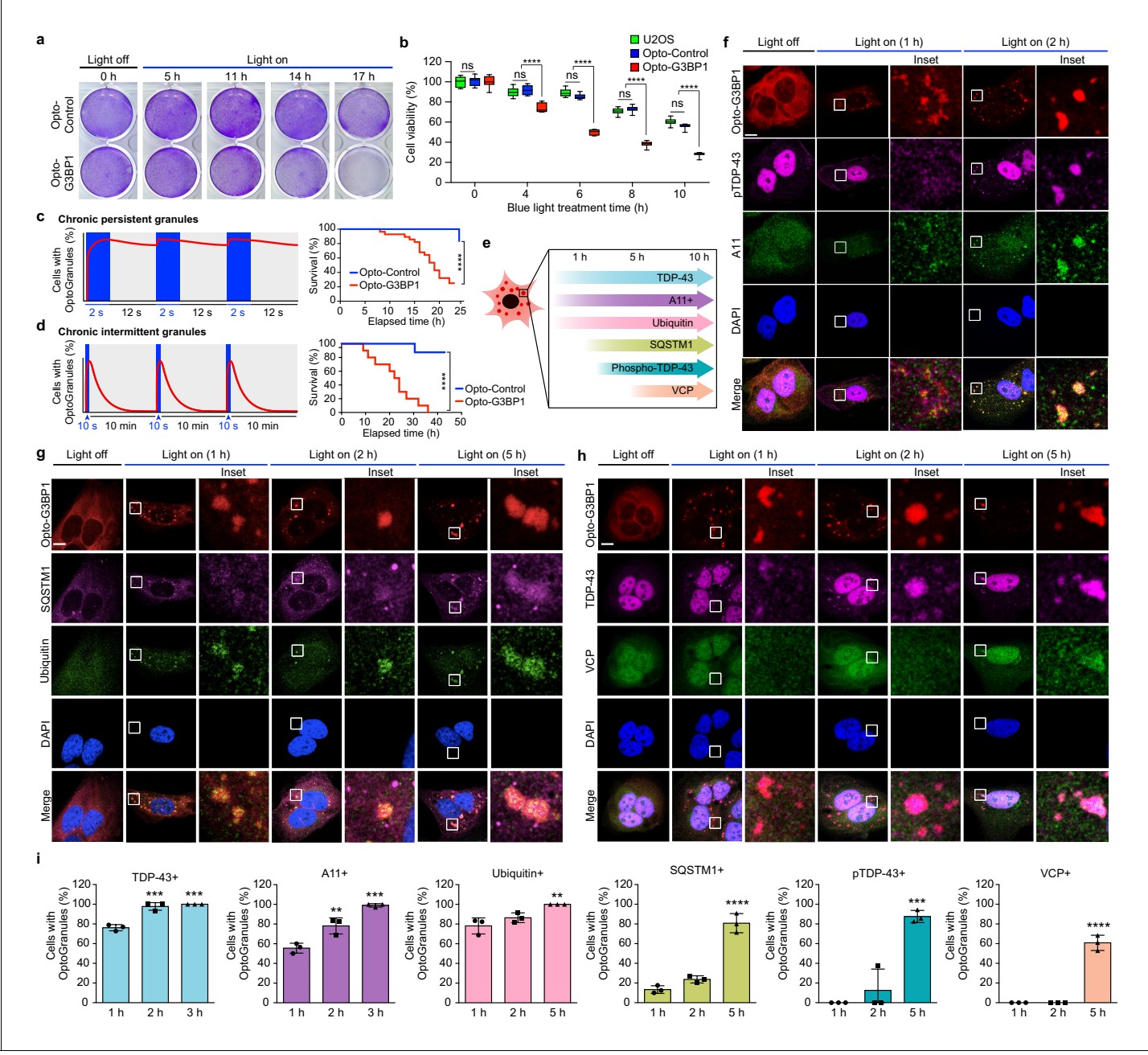

**Figure 3.** Persistent OptoGranules are cytotoxic and evolve to pathological inclusions. (**a,b**) U2OS cells stably expressing Opto-Control or Opto-G3BP1 were stimulated with continuous blue light (~2 mW/cm²) for indicated times using custom-made LED arrays and viability was assessed by crystal violet staining (**a**) or CellTiter-Glo 2.0 luminence (**b**). Whiskers represent minimum to maximum from n = 9 biological replicates. ****p<0.0001.; ns, not significant by two-way ANOVA with Tukey's post-test. (**c,d**) U2OS cells stably expressing Opto-Control or Opto-G3BP1 were exposed to chronic persistent (**c**) or chronic intermittent (**d**) blue light (445 nm) stimulation with live-cell imaging (power density ~0.12 W/cm²) as illustrated in the schematic (left) and assessed for cell survival by counting living cells (right). Blue boxes in schematic indicate the timing of light induction; red line is an idealized graph of the cellular response. Chronic persistent paradigm: n = 26 for Opto-Control and n = 28 for Opto-G3BP1. Chronic intermittent paradigm: n = 7 for Opto-Control and n = 10 for Opto-G3BP1. Data are shown from n = 3 independent experiments. ****p<0.0001 by log-rank (Mantel-Cox) test. (**e**) Timeline of protein accumulation in OptoGranules in U2OS cells. (**f-h**) U2OS cells stably expressing Opto-G3BP1 were stimulated with continuous blue light (~2 mW/cm²) for indicated times using custom-made LED arrays and co-immunostained with p-TDP-43 and A11 antibodies (**f**), SQSTM1 and ubiquitin antibodies (**g**), or VCP and TDP-43 antibodies (**h**). (**i**) quantification of data from (**f-h**). Error bars represent s.e.m. Images in f-h are representative of n = 3 independent experiments. ***p=0.0002 (2 hr), ***p=0.0001 (3 hr) for TDP-43, **p=0.0048 (2 hr), ***p=0.0002 (3 hr) for A11, **p=0.0051 (5 hr) for ubiquitin, ****p<0.0001 for SQSTM1, ***p=0.0003 for pTDP-43, and ****p<0.0001 for VCP by one-way ANOVA with Dunnett's test. Scale bars, 10 µm in all micrographs.

*Figure 3 continued on next page*

*Figure 3 continued*

DOI: https://doi.org/10.7554/eLife.39578.012

The following figure supplement is available for figure 3:

**Figure supplement 1.** OptoGranules evolve to pathological inclusions.

DOI: https://doi.org/10.7554/eLife.39578.013

oligomer, a feature also shared by conventional stress granules (*Figure 3e–i* and *Figure 3—figure supplement 1b*). The presence of A11 immunopositivity in newly formed stress granules suggests that non-pathological amyloid oligomers are present in the mRNPs recruited to these structures, perhaps arising from the prion-like low complexity domains of RNA-binding proteins coating these mRNPs. While these are presumably physiological amyloids, it is conceivable that their close packing in the condensed liquid state of persistent stress granules risks seeding the assembly of pathological amyloids, particularly for proteins like TDP-43 that can adopt highly stable structures.

Remarkably, the characteristics of OptoGranules changed during chronic assembly. First, OptoGranules showed time-dependent reduction in dynamics as assessed by FRAP (*Figure 3—figure supplement 1c,d*). Moreover, we observed that immunopositivity for TDP-43, A11, and ubiquitin gradually increased over time, and after approximately two hours of OptoGranule assembly, these granules showed immunopositivity using two distinct anti-phospho-TDP-43 antibodies (*Figure 3f–i*, *Figure 3—figure supplement 1e,f*). The anti-phospho-TDP-43 antibodies specifically recognize phosphorylation of TDP-43 at residues Ser409/410, a pathological signature specific to a spectrum of sporadic and familial forms of TDP-43 proteinopathies, including ALS-FTD (*Neumann et al., 2009*). Moreover, after approximately five hours of chronic OptoGranule assembly, we observed a significant increase in immunopositivity using antibodies to phospho-TDP-43 and the ubiquitin-binding proteins SQSTM1 and VCP, illustrating further evolution of these structures (*Figure 3g–i*).Thus, not only does chronic OptoGranule assembly cause a loss of cell viability, but cell death is preceded by the evolution of OptoGranules into cytoplasmic inclusions that recapitulate features that are pathognomonic for ALS-FTD.

We next examined the consequence of protracted stress granule assembly in a more disease-relevant, neuronal context by generating human induced pluripotent stem cell (iPSC)-derived neurons, which we verified had a cortical molecular identity (*Figure 4—figure supplement 1a,b*). In response to arsenite or heat shock stresses, these iPSC-derived neurons assembled conventional stress granules that were positive for TIA1 and TDP-43, indicating that they were suitable for examining the consequences of chronic stress granules (*Figure 4—figure supplement 1c*). Next, we introduced mRuby-tagged Opto-G3BP1 into differentiated neurons (*Figure 4a*). In mRuby-Opto-G3BP1-expressing neurons, blue light activation induced the assembly of OptoGranules indistinguishable from those observed in U2OS cells (*Figure 4b*, *Figure 4—figure supplement 1d* and *Videos 4* and *5*). Chronic induction of OptoGranules following transient introduction of mRuby-Opto-G3BP1 resulted in progressive loss of neuronal viability (*Figure 4c*) and the formation of neuronal cytoplasmic inclusions that were immunopositive for TDP-43, A11, and ubiquitin, with time-dependent immunopositivity for phosphorylated TDP-43 and SQSTM1 (*Figure 4d–i*, *Figure 4—figure supplement 1e*). We also generated iPSCs stably expressing inducible Opto-G3BP1 (doxycycline-inducible mCherry-tagged Opto-G3BP1), in which Opto-G3BP1 expression was induced simultaneously with neuronal differentiation by the addition of doxycycline (*Figure 4—figure supplement 2a*). In these cells, Opto-G3BP1 remained diffuse until activation with blue light, whereupon these neurons assembled dynamic OptoGranules (*Figure 4—figure supplement 2b*). With continuous stimulation, these OptoGranules further evolved into neuronal cytoplasmic inclusions that were positive for phospho-TDP-43, A11, ubiquitin, and SQSTM1 (*Figure 4—figure supplement 2c–f*). Thus, chronic OptoGranule induction recapitulates the evolution of ALS-FTD pathology and neurotoxicity in human iPSC-derived neurons.

## Discussion

In addition to providing a system to experimentally examine previously untestable hypotheses regarding the role of stress granules in disease, the development of the OptoGranule system

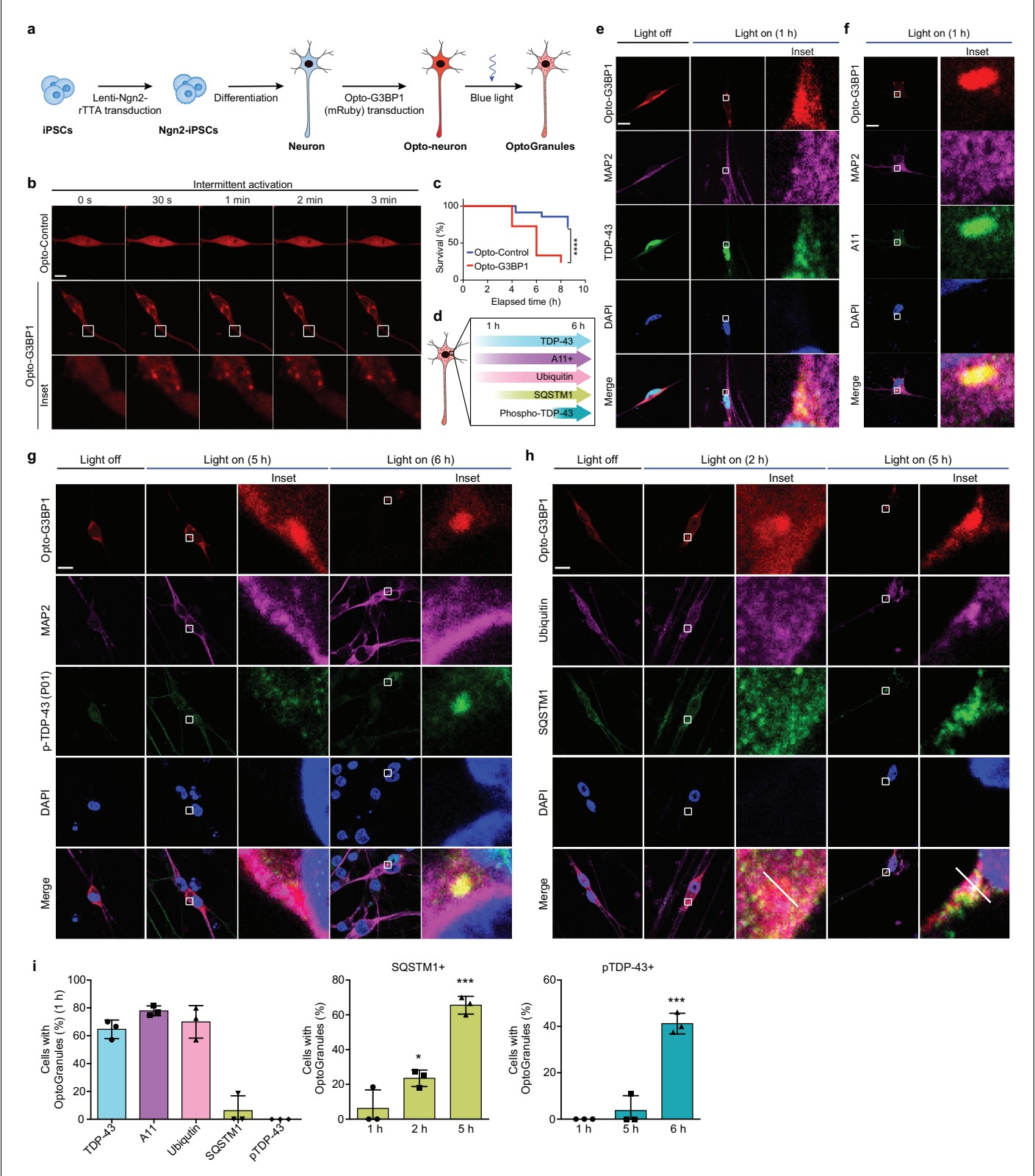

**Figure 4.** Persistent OptoGranules are cytotoxic and evolve to pathological inclusions in human iPSC-derived neurons. (**a**) Schematic illustrating generation of iPSC-derived neurons stably expressing Opto-G3BP1. (**b**) iPSC-derived neurons expressing Opto-Control (mRuby) or Opto-G3BP1 (mRuby) were intermittently exposed to a 488 nm blue-light laser (90% laser power, power density 6.3 W/cm²) followed by image acquisition with a 561 nm channel. Representative images are shown from n = 3 independent experiments. (**c**) iPSC-derived neurons expressing Opto-Control or Opto-
*Figure 4 continued on next page*

*Figure 4 continued*

G3BP1 were exposed to chronic persistent stimulation as in *Figure 3c* and survival was assessed by counting living cells. n = 35 cells for Opto-Control and n = 34 cells for Opto-G3BP1. Data are representative of n = 3 independent experiments. ****p<0.0001 by log-rank (Mantel-Cox) test. (**d**) Timeline of pathological protein accumulation in OptoGranules in iPSC-derived neurons. (**e-h**) iPSC-derived neurons expressing Opto-G3BP1 were stimulated with continuous blue light (~2 mW/cm$^2$) for indicated times using custom-made LED arrays and co-immunostained with MAP2 and TDP-43 antibodies (**e**), MAP2 and A11 antibodies (**f**), MAP2 and p-TDP-43 (P01) antibodies (**g**), or ubiquitin and SQSTM1 antibodies (**h**). See also *Figure 4—figure supplement 1e* for line scans of images shown in (**h**).(**i**) quantification of data from e-h. Error bars represent s.e.m. Images in e-h are representative of n = 3 independent experiments. *p=0.0489 (2 hr), ***p=0.0001 (5 hr) for SQSTM1 and ****p<0.0001 for pTDP-43 by one-way ANOVA with Dunnett's test. Scale bars, 10 μm in all micrographs.

DOI: https://doi.org/10.7554/eLife.39578.014

The following figure supplements are available for figure 4:

**Figure supplement 1.** iPSC-derived neurons form OptoGranules.
DOI: https://doi.org/10.7554/eLife.39578.015

**Figure supplement 2.** OptoGranules evolve to pathological inclusions in iPSC-derived neurons.
DOI: https://doi.org/10.7554/eLife.39578.016

provides insights into the nucleation and assembly of stress granules. In particular, we contrast the consequences of optogenetically enforced, intracellular LLPS of G3BP1 to those of FUS, TDP-43, and TIA1. These proteins are all constituents of stress granules (*Jain et al., 2016*) that undergo LLPS in vitro (Yang, Mathieu *et al.*, unpublished) (*Mackenzie et al., 2017*; *Patel et al., 2015*; *Conicella et al., 2016*). Yet, LLPS of G3BP1 results in the formation of OptoGranules, whereas FUS, TDP-43, and TIA1 form OptoDroplets that do not initiate stress granule assembly. OptoGranules and OptoDroplets are similar insofar as both types of assemblies are initiated with optically induced LLPS. Indeed, it is this similarity that suggested the name 'OptoGranules,' since they were inspired by and built upon observations made by Shin *et al.* regarding OptoDroplets. Beyond this similarity, however, OptoDroplets and OptoGranules are fundamentally different. This distinction is straightforward when considering evidence that biomolecular condensates are composed of clients and scaf-folds that play fundamentally different roles in the assembly and maintenance of these condensates (*Banani et al., 2016*). In unpublished work that strongly informed the development of OptoGranules, we identified G3BP as a uniquely essential central scaffold protein for stress granules, in contrast to TIA1, TDP-43, FUS and many others, which are client proteins (Yang, Mathieu *et al.*, unpublished). The differences in the assemblies formed by client proteins versus those formed by scaffold proteins make sense, since client proteins often reside in multiple biomolecular condensates with distinct identities. In contrast, enforced LLPS of a scaffold protein initiates the cascade of events that seeds the assembly of a full-fledged, complex stress granule.

We also highlight a second, more subtle distinction. OptoDroplets formed by Opto-TDP-43, Opto-FUS, and Opto-TIA1 have their biophysical origin in CRY2 oligomerization that presumably forces the IDRs of these proteins to self-associate and initiate a phase transition (*Shin et al., 2017*). In contrast, activation of Opto-G3BP1 forms granules because CRY2-based multimerization (specifically via the replaced NTF2L domain) increases the valency of G3BP, permitting it to engage with another scaffolding

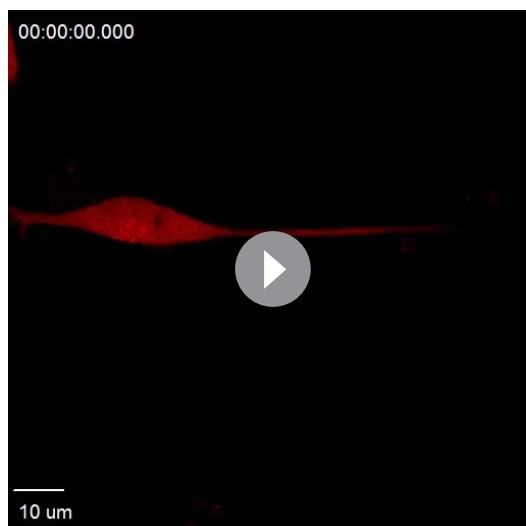

**Video 4.** Blue light activation fails to induce the assembly of OptoGranules in iPSC-derived neurons expressing Opto-Control. iPSC-derived neurons expressing Opto-Control (mRuby) were intermittently exposed to a 488 nm blue-light laser (95% laser power, power density 6.5 W/cm$^2$) followed by image acquisition with a 561 nm channel. See *Video 5* for corresponding Opto-G3BP1 condition.
DOI: https://doi.org/10.7554/eLife.39578.017

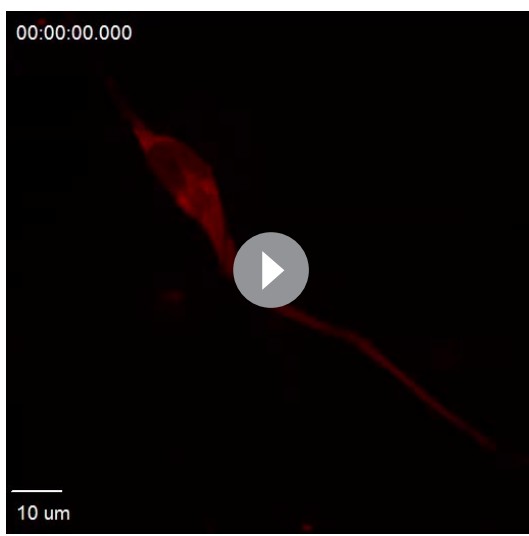

**Video 5.** Blue light activation induces the assembly of OptoGranules in iPSC-derived neurons expressing Opto-G3BP1. iPSC-derived neurons expressing Opto-G3BP1 (mRuby) were intermittently exposed to a 488 nm blue-light laser (95% laser power, power density 6.5 W/cm$^2$) followed by image acquisition with a 561 nm channel. See *Video 4* for corresponding Opto-Control condition.
DOI: https://doi.org/10.7554/eLife.39578.018

element (i.e., a class of RNAs), and these interactions create a seed that subsequently undergoes a phase transition that mediates subsequent further assembly of a stress granule.

Among the many membrane-less organelles that arise through phase transitions, stress granules have drawn the most attention from the ALS-FTD field because of their cytoplasmic location, which matches the location of pathological deposits in ALS-FTD, and the many disease-associated proteins that are components of stress granules. However, we must emphatically note that LLPS-mediated assembly, dynamics, and material properties of stress granules must be viewed within the context of a larger cellular network of membrane-less organelles, which include a wide variety of nuclear and cytoplasmic RNP granules. Indeed, membrane-less organelles are now recognized as functionally relevant biomolecular condensates that underlie different segregated biochemistries within a single cell (*Banani et al., 2017*). Furthermore, their material properties (e.g., assembly/disassembly rates, mobility, viscosity) likely influence these functions; indeed, the data presented here supports the burgeoning hypothesis that ALS-FTD arises from disturbances in the dynamics and material properties of membrane-less organelles, with devastating consequences over time.

Extending this hypothesis, we speculate that disease may reflect simultaneous pathological disturbance of multiple membrane-less organelles that arises by derangement of a network of multiple, independent phases. These interconnections likely reflect communication across different types of membrane-less compartments based on rapid, dynamic exchange of macromolecules (e.g., RNA and RNA-binding proteins) and small molecules that act as vehicles to communicate material states throughout the network. An example of this is seen in the recent report that perturbation of one phase-separated compartment (stress granules) alters the properties and function of a distinct phase-separated structure (the nuclear pore) (*Zhang et al., 2018*). With such a system-wide regulation, primary disturbances in the material properties of one node of the network (e.g., stress granules) may lead to secondary disturbances that are propagated throughout the entire network of membrane-less organelles.

## Materials and methods

### Key resources table

| Reagent type (species) or resource | Designation | Source or reference | Identifiers | Additional information |
|---|---|---|---|---|
| Cell line (Human) | U-2 OS | ATCC | HTB-96; RRID:CVCL_0042 | |
| Cell line (Human) | Lenti-X 293T(293LE) | Clontech | 632180; RRID: CVCL_4401 | |
| Cell line (Human) | iPSCs | Building a Kidney | BJFF6; RRID: CVCL_VU02 | |
| Recombinant DNA reagent | pCRY2PHR-mCherryN1 | Addgene | 26866; RRID:Addgene_26866 | |

*Continued on next page*

*Continued*

| Reagent type (species) or resource | Designation | Source or reference | Identifiers | Additional information |
|---|---|---|---|---|
| Recombinant DNA reagent | pCMV-CRY2-mCherry | Addgene | 58368; RRID:Addgene_58368 | |
| Recombinant DNA reagent | phND2-N174 | Addgene | 31822; RRID:Addgene_31822 | |
| Recombinant DNA reagent | pKanCMV-mClover3-mRuby3 | Addgene | 74252; RRID:Addgene_74252 | |
| Recombinant DNA reagent | pTight-hND2-N106 | Addgene | 31875; RRID:Addgene_31875 | |
| Recombinant DNA reagent | psPAX2 | Addgene | 12260; RRID:Addgene_12260 | |
| Recombinant DNA reagent | CRY2olig-mCherry | Addgene | 60032; RRID:Addgene_60032 | |
| Recombinant DNA reagent | pMD2.G | Addgene | 12259; RRID:Addgene_12259 | |
| Recombinant DNA reagent | linear hygromycin marker | Clontech | 631625; RRID:Addgene_60032 | |
| Antibody | goat polyclonal anti-β-actin | Santa Cruz Biotechnology | sc-1616; RRID: AB_630836 | (1:1000) |
| Antibody | mouse monoclonal anti-eIF2α | Santa Cruz Biotechnology | sc-133132; RRID: AB_1562699 | (1:1000) |
| Antibody | rabbit polyclonal anti-β-actin | Cell Signaling | 3597S; RRID: AB_390740 | (1:1000) |
| Antibody | rabbit polyclonal anti-mCherry | Abcam | 167453; RRID: AB_2571870 | (1:1000) |
| Antibody | mouse monoclonal anti-G3BP1 | BD Biosciences | 611126; RRID: AB_398437 | (1:1000) |
| Antibody | rabbit polyclonal anti-PABP | Abcam | ab21060; RRID: AB_777008 | (1:400) |
| Antibody | rabbit polyclonal anti-eIF4G | Santa Cruz Biotechnology | sc-11373; RRID: AB_2095750 | (1:400) |
| Antibody | rabbit polyclonal anti-TDP-43 | Proteintech | 12892–1-AP; RRID: AB_2200505 | (1:400) |
| Antibody | mouse monoclonal anti-phospho-TDP-43 (M01) | Cosmo Bio CO | TIP-PTD-MO1; RRID: AB_1961900 | (1:1000) |
| Antibody | rabbit polyclonal anti-phospho-TDP-43 (P01) | Cosmo Bio CO | TIP-PTD-PO1; RRID: AB_1961899 | (1:400) |
| Antibody | mouse monoclonal anti-VCP | BD Biosciences | 612183; RRID: AB_399554 | (1:100) |
| Antibody | rabbit polyclonal anti-amyloid-oligomer A11 | Thermo Fisher Scientific | AHB0052; RRID: AB_2536236 | (1:100) |
| Antibody | rabbit polyclonal anti-Ubiquitin | Dako | Z0458; RRID: AB_2315524 | (1:100) |

*Continued on next page*

*Continued*

| Reagent type (species) or resource | Designation | Source or reference | Identifiers | Additional information |
|---|---|---|---|---|
| Antibody | mouse monoclonal anti-p62 | Abcam | ab56416; RRID: AB_945626 | (1:400) |
| Antibody | mouse monoclonal anti-MAP2 | Sigma | M9942; RRID: AB_477256 | (1:400) |
| Antibody | goat polyclonal anti-TIA1 | Santa Cruz Biotechnology | sc-1751; RRID: AB_2201433 | (1:400) |
| Antibody | mouse monoclonal anti-TIAR | BD Biosciences | 610352; RRID: AB_397742 | (1:400) |
| Antibody | rabbit polyclonal anti-ataxin 2 | Proteintech | 21776–1-AP; RRID: AB_10858483 | (1:400) |
| Antibody | goat polyclonal anti-eIF3η | Santa Cruz Biotechnology | sc-16377; RRID: AB_671941 | (1:400) |
| Antibody | rabbit polyclonal anti-GLE1 | Abcam | ab96007; RRID: AB_10678755 | (1:400) |
| Antibody | rabbit polyclonal anti-FUS | Bethyl Laboratories | A300-302A; RRID: AB_309445 | (1:400) |
| Antibody | rabbit polyclonal anti-OPTN | Proteintech | 10837–1-AP; RRID: AB_2156665 | (1:400) |
| Commercial assay or kit | FuGENE 6 | Promega | E2691 | |
| Commercial assay or kit | NEBuilder HiFi DNA Assembly Master Mix kit | NEB | E2621 | |
| Commercial assay or kit | Q5 site-directed mutagenesis | NEB | E0054 | |
| Commercial assay or kit | RNA 3' End Biotinylation Kit | Pierce | 20160 | |
| Commercial assay or kit | CellTiter-Glo 2.0 assay kit | Promega | G9242 | |
| Chemical compound, drug | ISRIB | Sigma | SML0843 | 200 nM |
| Chemical compound, drug | cycloheximide | Sigma | C4859 | 100 µg/ml |
| Chemical compound, drug | sodium arsenite | Sigma | 35000–1 L-R | 0.5 mM |
| Chemical compound, drug | hygromycin B | Thermo Fisher Scientific | 10687010 | 200 µg/ml |
| Chemical compound, drug | doxycycline hyclate | Sigma-Aldrich | D9891 | 1 µg/ml |
| Chemical compound, drug | puromycin | Thermo Fisher Scientific | A1113803 | 1 µg/ml |
| Software, algorithm | ImageJ | NIH | https://imagej.nih.gov/ij/, RRID:SCR_003073 | |
| Software, algorithm | GraphPad Prism | GraphPad Software Inc | http://www.graphpad.com/scientific-software/prism/ RRID:SCR_002798 | |

*Continued on next page*

*Continued*

| Reagent type (species) or resource | Designation | Source or reference | Identifiers | Additional information |
|---|---|---|---|---|
| Software, algorithm | SlideBook 6 | Intelligent Imaging Innovations | https://www.intelligent-imaging.com/slidebook.php RRID:SCR_014300 | |
| Software, algorithm | Image Studio | LI-COR | https://www.licor.com/bio/products/software/image_studio_lite/?utm_source=BIO+Blog&utm_medium=28Aug13post&utm_content=ISLite1&utm_campaign=ISLite, RRID: SCR_014211 RRID:SCR_015795 | |
| Software, algorithm | LAS X Software | Leica | https://www.leica-microsystems.com/products/confocal-microscopes/p/leica-tcs-sp8/ RRID:SCR_013673 | |

## Cell culture and transfection

U2OS cells were purchased from ATCC (HTB-96) and periodically authenticated by short tandem repeat (STR) profiling. U2OS cells were cultured in Dulbecco's modified Eagle's medium (HyClone) supplemented with 10% fetal bovine serum (HyClone SH30071.03 and SH30396.03), 1X GlutaMAX (Thermo Fisher Scientific 35050061), 50 U/ml penicillin, and 50 µg/ml streptomycin (Gibco 15140–122), and maintained at 37°C in a humidified incubator with 5% $CO_2$. FuGENE 6 (Promega E2691) was used for transient transfections per the manufacturer's instructions. *G3BP1/2* KO cells have been previously described (*Zhang et al., 2018*). U2OS cells stably expressing G3BP1-GFP have been previously described (*Figley et al., 2014*). Cells were checked for mycoplasma with MycoAlert Mycoplasma Detection Kit (Lonza LT07-318) and then regularly checked for mycoplasma by DAPI staining.

## Plasmids

DNA fragments encoding human G3BP1 and TIA1 were PCR-amplified from G3BP1 (DNASU HsCD00042033) and pEGFP-TIA1 (*Mackenzie et al., 2017*), respectively. FUS and TDP-43 were PCR-amplified from cDNA. The pCRY2PHR-mCherry backbone was PCR-amplified from pCRY2PHR-mCherryN1 (Opto-Control; Addgene 26866). DNA fragments encoding G3BP1, TIA1, TDP-43, or FUS were inserted into pCRY2PHR-mCherryN1 backbone using NEBuilder HiFi DNA Assembly Master Mix kit (NEB E2621). To create Opto-G3BP2, DNA fragments encoding G3BP2 were amplified from cDNA and inserted into pCMV-CRY2-mCherry (Addgene 58368) at XhoI and BamHI using NEBuilder HiFi DNA Assembly Master Mix. Mammalian codon-optimized pCRY2PHR-mCherry was PCR-amplified from pCMV-CRY2-mCherry. mRuby3 was PCR-amplified from pKanCMV-mClover3-mRuby3 (Addgene 74252). Opto-G3BP1 (mRuby) was assembled from codon-optimized pCRY2PHR-mCherry, mRuby3, and G3BP1 DNA using NEBuilder HiFi DNA Assembly Master Mix kit. Opto-G3BP1 (mRuby) lentiviral plasmids were constructed by inserting PCR-amplified CMV-promoted CRY2-mRuby-G3BP1 (ΔNTF2L) into PspXI and EcoRI linearized cloning backbone phND2-N174 (Addgene 31822) using NEBuilder HiFi DNA Assembly Master Mix kit. Dox-Opto-G3BP1 (mCherry) lentiviral plasmids were constructed by inserting PCR-amplified Opto-Control and Opto-G3BP1 into EcoRI-digested cloning backbone pTight-hND2-N106 (Addgene 31875) using NEBuilder HiFi DNA Assembly Master Mix kit. Truncations were introduced using Q5 site-directed mutagenesis (NEB E0054). G3BP1-GFP constructs have been previously described (*Lee et al., 2016*). All constructs were confirmed by sequencing.

## Drugs and heat shock treatments

ISRIB (200 nM; Sigma SML0843) and cycloheximide (100 µg/ml; Sigma C4859) treatment was performed for 30 min before adding sodium arsenite (0.5 mM; Sigma 35000–1 L-R) or blue light. For

sodium arsenite treatment, medium was changed to medium containing 0.25 mM or 0.5 mM sodium arsenite for 30 or 45 min as indicated in figure legends. For heat shock treatment, cells were transferred to a 42°C humidified incubator with 5% $CO_2$ for 1 hr.

## Lentivirus production

Lenti-X 293 T cells (293LE; Clontech 632180) were transfected at 80–90% confluency with viral vectors containing genes of interest and viral packaging plasmids psPAX2 (Addgene 12260) and pMD2. G (Addgene 12259) using polyethylenimine (Polysciences 24765–2). The medium was changed 24 hr after transfection. Viral supernatants were harvested 48 hr after transfection, filtered with 0.45 µM filters, and centrifuged at 100,000 x g at 4°C for 1.5 hr. Ultracentrifugation was carried out through a 20% (w/v in PBS) sucrose cushion at 100,000 x g at 4°C for 1.5 hr. Pellets were resuspended in 100 µl DMEM +10% FBS and stored at −80°C.

## Stable cell lines

Opto-Control (mCherry) or Opto-G3BP1 (mCherry) constructs were co-transfected with linear hygromycin marker (Clontech 631625) into U2OS cells using FuGENE 6 (Promega). 48 hr after transfection, 200 µg/ml hygromycin B (Thermo Fisher Scientific 10687010) was added to culture media for selection. mCherry-positive cells were selected using cell sorting to produce Opto-Control (mCherry) or Opto-G3BP1 (mCherry) stable cell lines. For efficient photoactivation, cells with high expression (top 10%) were selected using cell sorting. Filtered Opto-Control (mRuby) or Opto-G3BP1 (mRuby) viral supernatants and 8 µg/ml polybrene (Sigma H9268) were added to U2OS cells at ~50% confluency in 10 cm plates. mRuby-positive cells were selected using cell sorting to produce Opto-Control (mRuby) or Opto-G3BP1 (mRuby) stable cell lines.

## iPSC neuron differentiation

iPSC neurons were generated as described previously (*Zhang et al., 2013*) with modifications. iPSCs ((Re)Building a Kidney BJFF6) were dissociated with Gentle Cell Dissociation Reagent (Stemcell Technologies 07174) and 300,000 iPSCs were seeded into one Matrigel (Corning 354277)-coated well of a six-well plate in mTeSR medium (Stemcell Technologies 85850) containing 10 µM ROCK inhibitor (Stemcell Technologies 72302). The next day, the medium was changed to mTeSR medium.

To generate neurons expressing mRuby-tagged Opto-G3BP1 or Opto-Control, lentiviruses encoding Ngn2 and rTTA were added to the medium at MOI = 4, respectively, in the presence of hexadimethrine bromide (4 µg/ml; Sigma-Aldrich H9268) and the medium was changed 24 hr after transduction. When transduced iPSCs reached 75% confluency, 1 µg/ml of doxycycline hyclate (Sigma-Aldrich D9891) was added to mTeSR medium to induce Ngn2 expression. At day 2 of induction, iPSCs were dissociated with Gentle Cell Dissociation Reagent and 150,000 cells were seeded onto coverslips in one well of a 24-well plate or 4-well Nunc Lab-Tek chambered coverglass (Thermo Fisher Scientific 155382) coated with Poly-L-ornithine/laminin/fibronectin (Sigma-Aldrich P4957; Sigma-Aldrich L2020; Sigma-Aldrich F4759 (*Richner et al., 2015*)), and cultured in BrainPhys neuronal medium (Stemcell Technologies 05790) containing 1 × N2 (Thermo Fisher Scientific 17502048), 1 × B27 (Thermo Fisher Scientific 12587010), 20 ng/ml BDNF (Peprotech 450–02), 20 ng/ml GDNF (Peprotech 450–10), 500 µg/ml Dibutyryl cyclic-AMP (Sigma-Aldrich D0627), 200 nM L-ascorbic acid (Sigma-Aldrich A0278), 1 µg/ml natural mouse laminin (Thermo Fisher Scientific 23017–015), 1 µg/ml doxycycline, and 1 µg/ml puromycin (Thermo Fisher Scientific A1113803). Opto-Control (mRuby) or Opto-G3BP1 (mRuby) lentiviruses and 4 µg/ml hexadimethrine bromide (Sigma H9268) were added to iPSC neurons at 3–5 DIV. Media was changed approximately 12 hr after transduction and then half-changed every other day until the assay was performed.

To generate doxycycline-inducible iPSC-derived neurons with inducible expression of mCherry-tagged Opto-Control or Opto-G3BP1, lentiviruses encoding Ngn2, rTTA, and Dox-Opto-Control or Dox-Opto-G3BP1 were added to the medium at ~MOI = 4, respectively, in the presence of hexadimethrine bromide (4 µg/ml), and the medium was changed 24 hr after transduction. iPSCs were dissociated with Gentle Cell Dissociation Reagent and 150,000 cells were seeded into coverslips in one well of a 24-well plate or 4-well Nunc Lab-Tek chambered coverglass coated with Matrigel/Poly-L-ornithine/laminin/fibronectin and cultured in BrainPhys neuronal medium for 7 days. iPSC neuron

cultures were maintained in BrainPhys neuronal medium and half-changed every other day until the assay was performed.

## Blue-light LED treatment

Cells (30–60% confluency) were transferred into blue light illumination at ~2 mW/cm$^2$ using custom-made LED arrays in a humidified incubator with 5% CO$_2$ with blue-light LED array for continuous blue light stimulation. Custom-made LED arrays were arranged with a flexible LED strip light (Ustellar). The light intensity of LED arrays was measured by a power meter (ThorLabs S170C).

## Live-cell imaging

All live-cell imaging experiments were performed using a Marianas 2 spinning disk confocal imaging system except overnight images of cell viability assays (described below). Images were acquired using a 63×/1.4 Plan Apochromat objective. Cells were plated in 4-well Nunc Lab-Tek chambered coverglass (Thermo Fisher Scientific 155382). Before imaging, the medium was changed to Fluoro-Brite DMEM medium (Thermo Fisher Scientific A1896701) with 10% fetal bovine serum and 1X GlutaMAX. During imaging, cells were maintained at 37°C with an environmental control chamber. Definite focus was used during the live-cell imaging. For one-time photoactivation, indicated cells were initially photoactivated by a 5 ms pulse of 488 nm laser illumination at 55% of maximum laser power, then imaged every 1 s thereafter with a 561 nm laser. For intermittent activation, cells were intermittently exposed to a 488 nm blue light (100 ms, 90% laser power, power density 6.3 W/cm$^2$) followed by image acquisition with a 561 nm channel. Images were analyzed with SlideBook 6 software. Laser power intensity was measured by a power meter (ThorLabs S170C).

## Western blotting

Cells were collected using PBS and lysed for 10 min on ice using RIPA buffer (25 mM Tris-HCl (pH 7.6), 150 mM NaCl, 1% NP-40, 1% sodium deoxycholate, 0.1% SDS; Pierce, 89901) supplemented with proteinase inhibitor cocktail (Roche 1186153001) and PhosSTOP (Roche 04906845001). Samples were centrifuged for 20 min at 4°C at 14,000 rpm. 4X NuPAGE LDS sample buffer (Thermo Fisher Scientific NP0008) was added to the supernatant and samples were boiled for 5 min. Samples were run in 4–12% NuPAGE Bis-Tris gels (Invitrogen) and transferred to nitrocellulose membranes using an iBlot 2 transfer device (Thermo Fisher Scientific). Membranes were blocked with Odyssey blocking buffer (LI-COR) and then incubated with primary antibodies. Following incubation with dye-labeled secondary antibodies, signals were visualized using an Odyssey Fc imaging system (LI-COR). Primary western blot antibodies were anti-β-actin (Santa Cruz Biotechnology sc-1616), anti-eIF2α (Santa Cruz Biotechnology sc-133132), anti-phospho-eIF2α (Cell Signaling 3597S), anti-mCherry (Abcam 167453), and anti-G3BP1 (BD biosciences 6111126). Secondary western blot antibodies were IRDye 800CW/680RD (LI-COR) used at a dilution of 1:15,000.

## Immunofluorescence

Cells were grown in 8-well chamber slides (Millipore). Following the indicated stimulation, cells were fixed with 4% paraformaldehyde (Electron Microscopy Science) in PBS for 10 min at room temperature, permeabilized with 0.2% Triton X-100 in PBS for 10 min at room temperature, and then blocked with 10% normal goat serum (Life Technologies 50062) or 5% BSA for 1 hr at room temperature. Samples were incubated with primary antibodies in blocking buffer overnight at 4°C. Samples were then washed three times with PBS and incubated with secondary antibody for 1 hr at room temperature. Primary antibodies were anti-PABP (Abcam ab21060), anti-G3BP1 (BD Biosciences 611126), anti-eIF4G (Santa Cruz Biotechnology sc-11373), anti-TDP-43 (Proteintech 12892–1-AP), anti-phospho-TDP-43 (M01) (Cosmo Bio CO TIP-PTD-MO1), anti-phospho-TDP-43 (P01) (Cosmo Bio CO TIP-PTD-PO1), anti-VCP (BD Biosciences 612183), anti-amyloid-oligomer A11 (Thermo Fisher Scientific AHB0052), anti-ubiquitin (Dako, Z0458), anti-SQSTM1 (Abcam ab56416), anti-MAP2 (Sigma M9942), anti-TIA1 (Santa Cruz Biotechnology sc-1751), anti-TIAR (BD Biosciences 610352), anti-eIF3η (Santa Cruz Biotechnology sc-16377), anti-ataxin 2 (Proteintech 21776–1-AP), anti-FUS (Bethyl Laboratories A300-302A), anti-OPTN (Proteintech 10837–1-AP), and anti-GLE1 (Abcam ab96007). Secondary antibodies were Alexa Fluor 488/555/647 (Life Technologies). For microscopic imaging,

slides were mounted with ProLong Gold Antifade Mountant with DAPI (Invitrogen). Images were captured using a Leica TCS SP8 3X confocal microscope with a 63x oil objective.

## Fluorescence recovery after photobleaching

Cells were first stimulated with a blue-light laser (power density ~4.5 W/cm$^2$) for 3 mins to initiate granule formation. Regions of interest expressing Opto-Control, Opto-Control (olig), Opto-G3BP1, or G3BP1-GFP were then photobleached and mCherry or GFP signal intensity was measured before and after photobleaching.

## Fluorescence in situ hybridization

Cells were fixed with 4% paraformaldehyde at room temperature for 10 min and then washed twice with PBS. 70% (v/v) EtOH was then added and cells were stored at 4°C overnight. Cells were then washed twice with wash buffer (2x SSC with 10% formamide in RNase-free water). Following aspiration of the wash buffer, cells were incubated with hybridization buffer (2x SSC, 10% v/v deionized formamide, 10% (w/v) dextran sulfate, 2 mM vanadyl ribonucleoside complex, 1 mg/ml yeast tRNA (Ambion AM7119), 0.005% BSA (Ambion AM2616) with 1 ng/µl 5′ labeled FAM-oligo(dT20) probes (Genelink 26-4620-02) at 37°C overnight. Cells were then washed 3 times with pre-warmed wash buffer at 37°C.

## Crystal violet assay

Cells were seeded at ~20% confluency in 6-well plates and grown for 24 hr before exposure to LED blue light. At the indicated treatment time, media was aspirated and replaced with staining solution (0.05% (w/v) crystal violet, 1% formaldehyde, 1% methanol in 1X PBS) for 20 min at room temperature followed by three washes with water.

## CellTiter-Glo 2.0 cell viability assay

This assay determines the number of viable cells by measuring ATP, which indicates the presence of metabolically active cells. Cells were seeded at 4–5 × 10$^3$ cells/well in 96-well plates one day before exposure to blue-light LED. Following blue light exposure, cells were measured using the CellTiter-Glo 2.0 assay kit (Promega G9242) per the manufacturer's instructions.

## Neuron viability imaging

Opto-Control and Opto-G3BP1 mRuby neurons were imaged on a DMI8 Widefield Microscope (Leica) with a 20x Plan Apo 0.80NA air objective using LAS X 3.4.2.18368 software (Leica). By saving the stage positions, a tilescan capture was taken in the same location every 2 hr using the 561 nm filter at 600 ms exposure. Between imaging, neurons were placed in the blue-light LED incubator until the next time point. Stitching was performed in LAS X, with each merged image totaling an area of ~37.82 mm$^2$.

## Overnight live-cell imaging

Overnight live-cell imaging experiments were performed with an Opterra II Swept Field confocal microscope (Bruker) using Prairie View 5.4 Software. Opto-Control and Opto-G3BP1 cells were plated in the middle two wells of a 4-well Lab-Tek chambered coverglass (Nunc) at ~20% confluency the day prior to imaging. Immediately before imaging, the medium was changed to FluoroBrite DMEM medium supplemented with 10% fetal bovine serum and 1X GlutaMAX. During imaging, cells were maintained at 37°C and supplied with 5% CO$_2$ using a Bold Line Cage Incubator (Okolabs) and an objective heater (Bioptechs). Imaging was performed using a 60x Plan Apo 1.40NA oil objective and Perfect Focus (Nikon) was engaged for the duration of the capture. Continuous activation data was acquired with a script made in Prairie View. The script was set to image the 561 nm channel with 100 ms exposure at 80 power in a multipoint capture once, followed by imaging the 445 nm channel with 2000 ms exposure at 200 power in a multipoint capture five times. This script was repeated continually for the duration of the experiment. Three fields of Opto-Control and Opto-G3BP1 cells, each with similar expression levels, were chosen per experiment. Analysis was performed using ImageJ.

## Droplet digital PCR

The QX200 droplet digital PCR (ddPCR) system (Bio-Rad) was used to measure gene expression levels in iPSCs and iPSC-derived neurons. The reaction was carried out in 20 µl emulsion PCR reactions that contain 20,000 droplets. Total RNAs were extracted by RNeasy Mini kit (Qiagen, 74104) and genomic DNA was removed by column by RNase-Free DNase (Qiagen, 79254). The ddPCR assay consisted of the following components: 1 × One-Step RT-ddPCR mix for probes (Bio-Rad, 1864021), forward primer (900 nM), reverse primer (900 nM), probe (FAM or HEX, 250 nM), nuclease-free water, and 5 ng RNA. All primers and probes were purchased from Thermo Fisher Scientific (*MAP2*, Hs00258900; *OCT4*, Hs04260367; *BRN2*, Hs00271595; *FOXG1*, Hs01850784; *SYN1*, Hs00199577) or Bio-Rad (*RPP30*, 10031228). Droplets were generated in a droplet generator (Bio-Rad) and PCR was performed in a C1000 Touch thermal cycler (Bio-Rad) according to the manufacturer's recommendation. After PCR, readout of positive versus negative droplets was performed using a QX200 droplet reader (Bio-Rad) and calculated by QuantaSoft software version 1.7.4.0917 (Bio-Rad).

## Statistical analysis

$p > 0.05$ was considered not significant. $*p \leq 0.05$, $**p < 0.01$, $***p < 0.001$, and $****p < 0.0001$ by two-tailed Student's t test, one-way ANOVA or two-way ANOVA with post-test as indicated in figure legends, or Log-rank (Mantel-Cox) test as appropriate. Statistical analyses were performed in GraphPad Prism or Excel.

## Acknowledgements

We thank Natalia Nedelsky for editorial assistance. We thank Anderson Kanagaraj for assistance with DNA construct preparation and Aaron Gitler (Stanford University) for providing phospho-TDP-43 antibodies. This work was supported by funding from the Howard Hughes Medical Institute, NIH grant R35 NS097974, ALS Association grant 18-IIA-419, and St. Jude Research Collaborative on the Biology of Membrane-less Organelles to JPT. JPT is a consultant for Third Rock Ventures.

## Additional information

### Competing interests

J Paul Taylor: Reviewing editor, *eLife,* and a consultant for Third Rock Ventures. The other authors declare that no competing interests exist.

### Funding

| Funder | Grant reference number | Author |
|---|---|---|
| Howard Hughes Medical Institute | | J Paul Taylor |
| National Institutes of Health | R35NS097974 | J Paul Taylor |
| ALS Association | 18-IIA-419 | J Paul Taylor |
| St. Jude Children's Research Hospital | | J Paul Taylor |

The funders had no role in study design, data collection and interpretation, or the decision to submit the work for publication.

### Author contributions

Peipei Zhang, Data curation, Formal analysis, Validation, Investigation, Visualization, Methodology, Writing—original draft, Writing—review and editing; Baochang Fan, Peiguo Yang, James Messing, Data curation, Formal analysis, Validation, Investigation, Methodology; Jamshid Temirov, Data curation, Formal analysis, Investigation, Methodology; Hong Joo Kim, Formal analysis, Visualization, Methodology, Writing—original draft, Project administration, Writing—review and editing; J Paul

Taylor, Conceptualization, Resources, Formal analysis, Supervision, Funding acquisition, Visualization, Methodology, Writing—original draft, Project administration, Writing—review and editing

### Author ORCIDs
Peipei Zhang (iD) http://orcid.org/0000-0003-1742-1680
Hong Joo Kim (iD) http://orcid.org/0000-0002-9157-1612
J Paul Taylor (iD) http://orcid.org/0000-0002-5794-0349

### Decision letter and Author response
Decision letter https://doi.org/10.7554/eLife.39578.021
Author response https://doi.org/10.7554/eLife.39578.022

## Additional files

### Supplementary files
• Transparent reporting form
DOI: https://doi.org/10.7554/eLife.39578.019

### Data availability
All data generated or analysed during this study are included in the manuscript and supporting files.

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
