## [Decision Letter]

[**Editorial note:** This article has been through an editorial process in which the authors decide how to respond to the issues raised during peer review. The Reviewing Editor's assessment is that all the issues have been addressed.]

Thank you for submitting your article "Persistent or repetitive assembly of OptoGranules is cytotoxic and reveals the evolution of stress granules to ALS-FTD pathology" for consideration by *eLife*. Your article has been reviewed by 3 peer reviewers, one of whom was the Reviewing Editor, and the evaluation has been overseen by a Senior Editor. The reviewers have opted to remain anonymous.

The Reviewing Editor has highlighted the concerns that require revision and/or responses, and we have included the separate reviews below for your consideration. If you have any questions, please do not hesitate to contact us.

Major concerns:

1) The dynamic recruitment of other stress granule components to OptoGranules should be characterized in detail. It is important to determine whether they are sequentially recruited to OptoGranules.

2) We strongly encourage the authors to include quantification data in the revised version: for example,% of cells with stress granules plotted over time after induction in Figure 2A and 2B should be included. It is hard to assess the degree of recruitment of different factors without any quantification (i.e. just from pictures). Figure 4 also suffers from this lack of quantification. Although images in panels e-h are representative of 3 independent experiments, a clear indication of the% of Optogranules which colocalise with pTDP-43, ubiquitin, P62 etc. and how this changes with time is crucial.

Separate reviews (please respond to each point):

*Reviewer #1:*

Liquid-liquid phase separation drives the assembly of stress granules (SGs) in response to a variety of exogenous stressors. Accumulation of SGs is associated with the pathogenesis of amyotrophic lateral sclerosis (ALS) and frontotemporal dementia (FTD). The causal relationship has not been firmly demonstrated. In this elegant study, Zhang et al. developed a light-inducible stress granule system, termed OptoGranules, that allows the authors to control the dynamics and composition of SGs in living cells. Using this system, the authors found that OptoGranules contain canonical stress granule components. Zhang et al. further showed that persistence of OptoGranules is cytotoxic, recapitulating the pathology of ALS-FTS. Overall, the findings are interesting. This manuscript is suitable for publication in *eLife* after appropriate revisions.

1) The dynamic recruitment of other stress granule components to OptoGranules should be characterized in more detail. It is important to determine whether they are sequentially recruited to OptoGranules.

2) The authors showed that the formation of FUS-, TDP-43- or TIA1-based OptoDroplets does not recapitulate the formation of stress granules. OptoDroplets do not colocalize with G3BP1-GFP and PABP. Do OptoDroplets contain other components of SGs such as VCP, p62 and OPTN? Mutations in these ALS disease genes cause accumulation of SGs. Does mutant FUS, TDP-43 or TIA1 induce SG formation in this system?

Minor Comments:

In Figure 1B, enlarged images showing the disassembly of granules could be presented. Discrete granules are hard to see over high expression background.

*Reviewer #2:*

The paper by Zhang et al. describes the formation of Optogranules induced by blue light. These Optogranules consist of G3BP linked to a light-sensitive CRY2/PHR domain. It is demonstrated that the Optogranules contain polyadenylated mRNAs and stress granule markers (in contrast to the Optodroplets formed based on FUS, TDP-43 and TIA1). Apart from the characterization of the Optogranules, repetitive stimulation of cells with blue light induces cell death. This also leads to a change in the composition of the Optogranules as, amongst others, phospho-TDP43 and ubiquitin stain positive in these Optogranules. In general, this is an interesting paper and most of the experiments are convincingly performed. However, there are a few important issues that need to be addressed and that will further strengthen the conclusions of the paper.

Major comments

– A major concern is that at least some of the data presented in this manuscript are very qualitative in nature. A lot of what is presented is based on co-stainings of a few cells and colocalisation in (part of) one cell. Despite the fact that these are most likely typical examples, one always worries that there could be a selection bias and that stainings are more likely to be selected if these are in accordance with what one expects. With the data provided, it is impossible to judge whether for instance the maturation of the droplets into droplets that are positive for phospho-TDP-43 and ubiquitin, as well as for other markers, is a valid observation. These changes are nicely shown in a schematic overview in Figure 3E and Figure 4D. However, it is not clear on how many cells/droplets this is based and whether there was a real time-dependent quantification underlying these data. Especially Figure 4 suffers from this lack of quantification. Although images in panels e-h are representative of 3 independent experiments, a clear indication of the% of Optogranules which colocalise with pTDP-43, ubiquitin, P62 etc. and how this changes with time is crucial.

– Stress granules are thought to be transient structures, forming rapidly in response to stress. Although the authors demonstrate that Optogranules form rapidly in response to blue light, in seconds or minutes (Figure 1B, Figure 1G), it is unclear why the authors switch from an approx. 5 min induction to a 6 h induction paradigm (Figure 1H to 1J) when demonstrating recruitment of known stress granule proteins. Are these other stress granule proteins only recruited in response to sustained assembly of opto-G3BP1? The authors should look at earlier time points. The same induction paradigm is also used in Figure 2.

- It is not so clear whether it is indeed the change over time of the composition of the Optogranules after repetitive stimulation with blue light that is responsible for the induction of the cell death. Some additional experiments should be performed to clarify this issue. Is the dynamics of the formation and disassembly of Optogranules changing over time? It should also be investigated whether there is an effect of repetitive stimulation of the cells on the physicochemical properties of the stress granules. How is fluorescence for instance recovering after photobleaching?

– The added value of the iPSC related experiments is not so clear. It should be clarified in more detail in which sense the Optogranules are different in these cells in comparison to the Optogranules in the U2OS cells. At present, the overall impression is that what is concluded from these experiments is very similar (identical) to what was observed in the U2OS cells. If so, one wonders how the described process could lead to selective neuronal death.

Minor comments:

– The quality of some of the figures seems suboptimal (could be due to the conversion to a pdf). One clear example is the first figure illustrating the formation of Optogranules. While it is clear in the movie, it is much less clear to observe the individual droplets in Figure 1B.

– It is indicated that the Optocontrol construct also forms a modest amount of nuclear and cytoplasmic clusters. It is not clear what the identity of these clusters is. Can this be determined and discussed in more detail?

– No Opto-control is provided for the experiment in Figure 1G.

– Figure 1—figure supplement 2: The length of time that cells were exposed to blue light is not stated. Was this the same 6 h induction used in Figure 1?

– Figure 2A and 2B, quantification would be more convincing. e.g.% of cells with stress granules plotted over time after induction. The experiment should also be controlled using the Opto-control construct.

– Figure 2C and D: It is concluded that the Optogranule assembly is independent of the regulation by eIF2alpha. This important observation should be discussed in more detail.

– Figure 2E: Not clear whether the same blot was reprobed or whether different blots are shown for phospho-eIF2alpha and eIF2alpha. Why is the eIF2alpha at the top (and not in the middle) of the part of the blot that is shown?

– All stress granule components that were studied colocalised with the Optogranules. One obvious candidate that is missing in this long list is FUS (Results section). Was FUS present in these Optogranules?

– Figure 3A: What is the effect of continuous blue light on the U2OS cells?

– Figure 3C and D: From the Kaplan-Meier plots one can estimate that approximately 10-15 cells are followed after induction of the Optogranule formation. How were these cells selected and how many were selected in the three different independent experiments.

– Figure 3C and D: Is there a statistical difference between the cell death induced by the induction of chronic persistent granules or after induction of chronic intermittent granules?

– Figure 3: On the Y-axis of panel c and d, the numbers are missing. It is also not clear whether the blue boxes are based on a quantification or whether these boxes represent what is expected (should be specified in the legend).

– Figure 4: It is unclear whether persistent induction of Optogranules causes increased phosphorylation of TDP-43 or simply recruitment of a population which is already present in the cell. The authors should quantify the total levels of phospho-TDP-43 before and after chronic induction of Optogranule assembly to determine whether levels are increased.

– More details should be provided on the nature/quality of the neurons that are differentiated from the iPSCs. Are these cortical neurons? How pure are these neuronal cultures? Are these neurons electrically active?

Additional data files and statistical comments:

It is not always clear whether the correct statistics is used. One example is Figure 3B. A Student t-test doesn't seem to be the correct way to evaluate whether these differences are statistically different. Moreover, as these are box plots, it seems unlikely that the variation is indicated as standard deviations (s.d.) as is indicated in the figure legend.

*Reviewer #3:*

Very interesting paper – extremely easy to read and parse. Important for field, enjoyed reading, lots of interesting insights.

Some comments:

* It is hard to assess the degree of recruitment of different factors without any quantification (i.e. just from pictures). For example, it is not clear to be that there is bona fide p62 recruitment in neurons (Figure 4H). If this were a normal review I would insist this must be quantified to some degree (e.g. pixel mutual information using an intensity threshold as a binary mask, given intensities are hard to interpret).

* There are three distinct modes of illumination: continuous (Figure 3A), chronic persistent (Figure 3C) and chronic intermittent (Figure 3D). It is difficult to determine which is used in which experiment. I would strongly suggest standardizing this and ensuring each experiment is clearly labelled regarding which regime is used. Without this interpreting the data (for the reader) becomes difficult.

* The OptoXXX language used is a little confusing. In the Shin et al. paper, OptoDroplets refers to the general system of a Cry2 domain fused to something:

"Here, we introduce an optogenetic platform that uses light to activate IDR-mediated phase transitions in living cells. We use this 'optoDroplet' system to study condensed phases driven by the IDRs of various RNP body proteins"

Here, OptoGranules refer to the physical cellular assemblies (to distinguish them from stress granules – which makes semantic sense) and OptoG3BP1 is an example of an OptoDroplet construct that contains G3BP1. The fact that the OptoDroplet work is not used to introduce the technology is odd. The results determined here are of significant biological relevance and interest, but the approach is fundamentally the OptoDroplet platform applied to a different protein. The fact that OptoDroplets are only introduced once the OptoGranule system has been described:

'These results contrast with observations made with light-induced aggregation of so-called "OptoDroplets," which nicely…'

Gives the impression that this is a different approach system, which is not the case (Figure 1A from this paper and 1A from Shin et al. are essentially interchangeable), and comes across as a little disingenuous. This was a general sentiment I heard from various places when the original preprint went up. It is surprising to me, because all of the exciting results pertain to the biology, not to the method, so it's unclear why the approach is introduced without basically saying "Here we extended the OptoDroplet system to examine the role of G3BP1 in nucleating stress granules…" or something to this effect.

Minor Comments:

* It would be useful to draw the schematic (Figure 1A) with the domains proportional to their number of amino acids

* Cry2+mCherry = ~840 amino acids (96 kDa), while NTF2 = 142 amino acids (16 kDa)

* This, of course, may be impractical, in which case simply including the regional size in number of amino acids would be sufficient

* Is the absence of SGs in response to the double G3BP1/2 knockout a true absence of stress granules, or an absence of assemblies with G3BP1/Tia1/PAB1 markers – have other markers been checked?

* "…insinuate into stress granules and other membrane-less organelles" – not sure insinuate has an appropriate meaning here? 'Infiltrate' perhaps?

* Figure 1—figure supplement 1B is the same as part of Figure 1A?

* "Opto-Control expression remained diffuse, with a modest amount of nuclear and cytoplasmic clusters (Figure 1B and Supplementary Video 1)." – the text is intriguing, though it wasn't obviously clear to me where these cytoplasmic clusters are in Figure 1B? But assuming you do see some level of clusters, why is an obligate dimer able to lead to any degree of cluster formation? I would have expected dimers to be far far below the resolution limit such that monomer/dimer Opto-Control should be equally diffuse

* When the OptoControl is off it appears to partition into the nucleus (Figure 1H, 1I, 1J) but when activated it is strongly excluded. Both monomeric and dimeric version should be larger than the threshold size for free permeability into the nucleus sans an NLS, so its not clear how/why light should be changing the distribution – some explanation of this would be useful (i.e. does this imply there are additional un-characterized responses to blue-light stimulation?).

* Are the expression levels of OptoG3BP1 comparable to endogenous G3BP1? I realize expression is clearly a tunable parameter as addressed in Figure 2, so it would be useful, as a reference, to mention the expression levels of endogenous protein here.

* "Expression of Opto-constructs using full-length TDP-43 or FUS [CRY2PHR-mCherry-TDP-43(FL); CRY2PHR-mCherry-TDP-43(FL)] did not produce cytoplasmic clusters with blue light activation" (minor point – I believe the second instance here should be CRY2PHR-mCherry-FUS(FL))

This is a very interesting result for two reasons

1) It’ not clear to me why FL-FUS OptoDroplets should not form assemblies, especially given recent reports that FL FUS assemblies much more readily than the LCD alone. Any thoughts on this in the discussion would, I think, be very useful. [1]

2) Recent work has shown that the NTD of TDP43 forms dimers that are necessary for phase separation in full-length TDP-43. In agreement with this, the CRY2 dimerization domain appears able to replace this N-terminal domain and give rise to droplets, HOWEVER, if the N-terminal domain is present (i.e. in CRY2PHR-mCherry-TDP-43(FL)) droplet formation is suppressed – any possible mechanism here would be useful to address – specifically, how does the ADDITION of a dimerization domain between CRY2 and the C-terminal portion of TPD43 suppress assembly?

* These questions are directly related to the comparison of other optoDroplet constructs

* Not sure how to interpret the Opto-G3BP1 vs G3BP1-GFP RNA binding data? Naively from the description of the RNA binding assay I would have expected 1) cross-linking to cause G3BP1 in stress granules to extensively cross-link with local RNAs, while under non-stress/non OptoGranule inducing conditions there would be minimal RNA cross-linking and 2) for Opto-G3BP1 and G3BP1-GFP to be similar. Instead, the data appear to show that you get extensive and essentially identical RNA binding under stressed and non-stressed conditions, and that Opto-G3BP1 and G3BP1-GFP have different RNA binding profiles. Similarly, not sure how to interpret the western blot associated with 2b – it would be useful to explicitly annotate what the bands are. This is almost certainly just a limitation of my understanding of the experimental setup, but it seems at least plausible others might also be confused…

* In the protein name SQSTM1 is used, while in Figure 3 p62 is used – this may confusing for people unfamiliar with the protein, so I'd suggest a consistent naming scheme

* I found the mCherry vs. mRuby optoG3BP1 section confusing on the first read, as it was initially unclear why both would be needed. I would suggest splitting this into a 'first we used mRuby-optoG3BP1…', then once this has been completed, explain why an inducible system is needed and then introduce the inducible mCherry variant.

* Why is the pTDP-43 recruitment to SGs in neurons so much weaker than in U2OS cells? In U2OS see reasonable recruitment just 1 hour in, with robust recruitment after 2 hours. In neurons after 5 hours (4g) there is basically no co-localization with optoG3BP1, and then after 6 robust co-localization is observed.

[1] Qamar, S., Wang, G., Randle, S.J., Ruggeri, F.S., Varela, J.A., Lin, J.Q., Phillips, E.C., Miyashita, A., Williams, D., Ströhl, F., et al. (2018). FUS Phase Separation Is Modulated by a Molecular Chaperone and Methylation of Arginine Cation-π Interactions. Cell 173, 720-734.e15.

[2] Wang, A., Conicella, A.E., Schmidt, H.B., Martin, E.W., Rhoads, S.N., Reeb, A.N., Nourse, A., Montero, D.R., Ryan, V.H., Rohatgi, R., et al. (2018). A single N‐terminal phosphomimic disrupts TDP‐43 polymerization, phase separation, and RNA splicing. EMBO J. e97452.

---

## [Author Response]

The Reviewing Editor has highlighted the concerns that require revision and/or responses, and we have included the separate reviews below for your consideration. If you have any questions, please do not hesitate to contact us.Major concerns:1) The dynamic recruitment of other stress granule components to OptoGranules should be characterized in detail. It is important to determine whether they are sequentially recruited to OptoGranules.2) We strongly encourage the authors to include quantification data in the revised version: for example,% of cells with stress granules plotted over time after induction in Figure 2A and 2B should be included. It is hard to assess the degree of recruitment of different factors without any quantification (i.e. just from pictures). Figure 4 also suffers from this lack of quantification. Although images in panels e-h are representative of 3 independent experiments, a clear indication of the% of Optogranules which colocalise with pTDP-43, ubiquitin, P62 etc. and how this changes with time is crucial.

We thank the reviewers for their constructive critique of this manuscript. The reviewers raised some excellent points, which we address below.

We also wish to highlight our answer to one question from the reviewers regarding the difference between OptoDroplets and OptoGranules. These are similar insofar as both types of assemblies are initiated with optically induced phase separation. Indeed, it is this similarity that suggested the name “OptoGranules,” since they were inspired by and built upon observations made by Shin et al. regarding OptoDroplets. Beyond this similarity, however, OptoDroplets and OptoGranules are fundamentally different. This distinction is straightforward when considering evidence that biomolecular condensates are composed of *clients* and *scaffolds* that play fundamentally different roles in the assembly and maintenance of these condensates. In unpublished work that strongly informed the development of OptoGranules, we identified G3BP as a *uniquely essential central scaffold protein* for stress granules, in contrast to TIA1, TDP-43, FUS, and many others, which are *client* proteins (Yang, Mathieu et al., in review). Please note, this manuscript has been made available for assessment by the editors and reviewers. In Yang, Mathieu et al., we show that G3BP dimerization and multivalent RNA binding drives phase separation with long, single- stranded mRNAs to form complexes that seed stress granule assembly. Moreover, this work revealed that the native dimerization element in G3BP could be functionally replaced by heterologous dimerization domains, which informed the design of Opto-G3BP1. In fact, simply tagging G3BP1 with CRY2 *does not* produce a fusion protein capable of efficiently producing stress granules in response to optical induction. In the present manuscript, we show that enforced phase separation of *client* proteins (TIA1, TDP-43, and FUS) using CRY2 generates OptoDroplets, which are relatively homogenous intracellular condensates, thereby confirming the results of Shin et al., but we also demonstrate that these are not stress granules. This makes sense since *client* proteins often reside in multiple biomolecular condensates with distinct identities. In contrast, enforced phase separation of a *scaffold* protein initiates the cascade of events that seeds the assembly of a full-fledged, complex stress granule, which we term an OptoGranule. Thus, an important takeaway from this work is the critical role of specific scaffolding elements in establishing and maintaining the identity of specific biomolecular condensates.

Separate reviews (please respond to each point):

Reviewer #1:

Liquid-liquid phase separation drives the assembly of stress granules (SGs) in response to a variety of exogenous stressors. Accumulation of SGs is associated with the pathogenesis of amyotrophic lateral sclerosis (ALS) and frontotemporal dementia (FTD). The causal relationship has not been firmly demonstrated. In this elegant study, Zhang et al. developed a light-inducible stress granule system, termed OptoGranules, that allows the authors to control the dynamics and composition of SGs in living cells. Using this system, the authors found that OptoGranules contain canonical stress granule components. Zhang et al. further showed that persistence of OptoGranules is cytotoxic, recapitulating the pathology of ALS-FTS. Overall, the findings are interesting. This manuscript is suitable for publication in eLife after appropriate revisions.1) The dynamic recruitment of other stress granule components to OptoGranules should be characterized in more detail. It is important to determine whether they are sequentially recruited to OptoGranules.

The manner in which individual constituents are recruited to stress granules remains poorly defined, although it is clear than many mRNAs and associated RNA-binding proteins are “recruited” simultaneously in bulk because they are part of mRNPs that constitute the primary content of stress granules. Other proteins likely participate in the initial phase transition that drives stress granule assembly, and thus are also “recruited” from the outset. Indeed, dynamic imaging in Figure 1G shows simultaneous recruitment of TIA1 and G3BP1 to stress granules presumably for this reason. Yet, there are >400 distinct protein constituents and >1800 distinct RNAs that have been reported in stress granules. As the reviewer suggests, some constituents are sequentially recruited (which we take to mean that there is a temporal delay in their recruitment after a microscopically visible stress granule is evident). For example, we have demonstrated delayed recruitment of the ubiquitin-binding proteins p62/SQSTM1 and VCP, and this is preceded by the appearance of ubiquitin signal in OptoGranules (Figures 3G-I and 4H-I). The clear implication of this finding is that a specific post-translational modification results in the subsequent recruitment of constituents that were not initially part of an RNP and we are anxious to learn how common this is. Chemical modifications of RNA or protein may be a general mechanism contributing to both delayed recruitment of constituents or time-dependent maturation of stress granules (as shown by reduced dynamics in mature stress granules in new Figure 3—figure supplement 1C-D). The OptoGranule system provides a tool to address these and many other fundamental questions about the molecular mechanism underlying RNP granule assembly in general, stress granule assembly in particular, and the relationship of stress granules to a wide variety of biological functions. These questions go beyond the scope of this initial report, which aims to illustrate (1) proof of concept that optically driven aggregation of a key scaffolding element can seed assembly of a complete membrane-less organelle, and (2) that protracted stress granule assembly is cytotoxic and recapitulates features of a specific neurodegenerative disease.

2) The authors showed that the formation of FUS-, TDP-43- or TIA1-based OptoDroplets does not recapitulate the formation of stress granules. OptoDroplets do not colocalize with G3BP1-GFP and PABP. Do OptoDroplets contain other components of SGs such as VCP, p62 and OPTN? Mutations in these ALS disease genes cause accumulation of SGs. Does mutant FUS, TDP-43 or TIA1 induce SG formation in this system?

No, as shown in new Figure 1—figure supplement 4D-E, J OptoDroplets formed from light-induced aggregation of TDP-43, FUS, or TIA1 do not recruit or colocalize with p62/SQSTM1, VCP, or OPTN, just as they do not recruit canonical stress granule constituents. This is an important point and speaks to a larger question addressed partially in this manuscript but more explicitly in a companion manuscript currently under review (Yang, Mathieu et al., in review). Please note, this manuscript has been made available to *eLife* for assessment by editors and reviewers. Quoting from this companion manuscript:

“RNP granules are complex structures composed of hundreds of RNAs and proteins. Although it has become evident as a general principle that RNP granule assembly likely reflects cooperative protein-protein, protein-RNA, and RNA-RNA interactions between the many constituents of these structures, the precise interactions required for specific types of RNP granules are almost entirely unknown. Moreover, the underlying mechanism(s) that permit distinct RNP granule identities to be established and maintained are largely enigmatic. For example, nuclear speckles and Cajal bodies can be clearly distinguished from one another by morphology, subcellular localization, constituents, and function. A similar distinction is evident for cytoplasmic P bodies and stress granules. Given that these biomolecular condensates share common principles of assembly, are built from highly similar (and sometimes overlapping) constituent RNA molecules and proteins, and remain in dynamic equilibrium with a shared milieu, how do these structures establish and maintain distinct identities?”

In brief, the answer to this question is that RNP granule assembly is initiated by a limited number of scaffolding molecules and cannot be recapitulated by enforced phase transition of any random constituent. For stress granules, a whole-genome screen identified G3BP as a uniquely essential scaffolding factor in initiating assembly and maintaining identity (Yang, Mathieu et al., in review). This scaffolding protein is distinct from other protein components of stress granules (e.g., TDP-43, FUS, TIA1), which we classify as “client” proteins. This is why optically induced phase separation of G3BP1, an essential scaffold for stress granules, initiates assembly of a complex organelle, whereas optically induced aggregation of other proteins or fragments, such as client proteins TDP-43, FUS, or TIA1, does not.

OptoDroplets (e.g., condensates formed by client proteins) also differ from OptoGranules (e.g., complex granules formed by scaffold proteins) also provide different advantages for experimental purposes. The former offers a generalizable approach to monitor the biophysical phase transition properties of proteins in cells and correlate those with behavior in reconstituted biochemical systems. The latter builds upon the experimental foundation of the OptoDroplet system to seed the formation of a specific complex membrane-less organelle by using optogenetic oligomerization of a specific, essential scaffolding protein. Thus, OptoGranules can be used to test specific hypotheses regarding the assembly, kinetics, and maturation of specific types of complex organelles.

Importantly, the OptoGranule system requires knowledge of both the *identity* of the critical scaffolding elements and *how* these elements contribute to assembly. In this case, we used our prior knowledge of G3BP1 and the mechanism of stress granule assembly to rationally design the seed for a stress granule. This approach is also likely to be generalizable as the identities of essential scaffolding proteins are identified for a variety of RNP granules.

To answer the reviewer’s related question regarding whether exogenous overexpression of mutant FUS, TDP-43, or TIA1 induces stress granule formation in this system, we show in new Figure 1—figure supplement 1B that overexpression of mutant FUS, TDP-43, or TIA1 induces stress granule formation in Opto-G3BP1 cell lines, as expected.

Minor Comments:In Figure 1B, enlarged images showing the disassembly of granules could be presented. Discrete granules are hard to see over high expression background.

This is a good suggestion. We have now added enlarged images to expanded Figure 1B.

Reviewer #2:

The paper by Zhang et al. describes the formation of Optogranules induced by blue light. These Optogranules consist of G3BP linked to a light-sensitive CRY2/PHR domain. It is demonstrated that the Optogranules contain polyadenylated mRNAs and stress granule markers (in contrast to the Optodroplets formed based on FUS, TDP-43 and TIA1). Apart from the characterization of the Optogranules, repetitive stimulation of cells with blue light induces cell death. This also leads to a change in the composition of the Optogranules as, amongst others, phospho-TDP43 and ubiquitin stain positive in these Optogranules. In general, this is an interesting paper and most of the experiments are convincingly performed. However, there are a few important issues that need to be addressed and that will further strengthen the conclusions of the paper.Major comments– A major concern is that at least some of the data presented in this manuscript are very qualitative in nature. A lot of what is presented is based on co-stainings of a few cells and colocalisation in (part of) one cell. Despite the fact that these are most likely typical examples, one always worries that there could be a selection bias and that stainings are more likely to be selected if these are in accordance with what one expects. With the data provided, it is impossible to judge whether for instance the maturation of the droplets into droplets that are positive for phospho-TDP-43 and ubiquitin, as well as for other markers, is a valid observation. These changes are nicely shown in a schematic overview in Figure 3E and Figure 4D. However, it is not clear on how many cells/droplets this is based and whether there was a real time-dependent quantification underlying these data. Especially Figure 4 suffers from this lack of quantification. Although images in panels e-h are representative of 3 independent experiments, a clear indication of the% of Optogranules which colocalise with pTDP-43, ubiquitin, P62 etc. and how this changes with time is crucial.

Yes, quantitative information was collected in the course of conducting the presented experiments. This quantitative data, with accompanying statistical analyses, has now been added to the figures. With reference to the reviewer’s specific comments, we call attention to new Figure 3I and Figure 4I, which show quantification of stress granule markers over time.

A brief, but not unimportant note: OptoGranules *do not* “consist of G3BP linked to a light- sensitive CRY2/PHR domain.” In fact, simply tagging G3BP1 with CRY2 does not produce a fusion protein capable of efficiently producing stress granules in response to optical induction. For the system to work, one must replace the native dimerization element in G3BP with an optically inducible multimerization element. Moreover, OptoGranules *are not* merely droplets of G3BP1. Rather, enforced phase separation of G3BP1 initiates a cascade of assembly that results in the formation of a full-fledged, complex RNP granule that is indistinguishable from an endogenous stress granule. This is because G3BP1 is the essential protein scaffold of stress granules.

– Stress granules are thought to be transient structures, forming rapidly in response to stress. Although the authors demonstrate that Optogranules form rapidly in response to blue light, in seconds or minutes (Figure 1B, Figure 1G), it is unclear why the authors switch from an approx. 5 min induction to a 6 h induction paradigm (Figure 1H to 1J) when demonstrating recruitment of known stress granule proteins. Are these other stress granule proteins only recruited in response to sustained assembly of opto-G3BP1? The authors should look at earlier time points. The same induction paradigm is also used in Figure 2.

We used two different blue light sources depending on the experimental design and readout. Specifically, for live imaging experiments, in which stress granule dynamics were monitored in real time, including experiments with intermittent activation, we used a photoactivation laser tuned to 455 or 488 nm, which has a powerful energy density (~2.5 MW/cm^2^for one-time activation or > ~4.5 W/cm^2^ for repeated activations) in a defined region of interest. However, this approach allows only a few cells to be examined at one time. For experiments requiring larger numbers of cells and therefore necessitating a broader field of illumination, we switched to a blue-light LED array (37°C, 5% CO_2_, in a humidified incubator), which had a much lower energy density (~2 mW/cm^2^) and induced OptoGranule formation over a longer time frame. This approach was used for experiments in which cells were fixed and stained for subsequent quantitative assessment. Importantly, in both experimental approaches the recruitment of stress granule components matched the kinetics of OptoGranule assembly. We point the reviewer to Figure 1G, Supplementary Video 3, Figures 3F-3H, and Figures 4E-4F, in which we assessed recruitment of stress granule markers at earlier time points (1-4 minutes and 1-2 hours post-induction) compared to the 6-hour time point shown in Figures 1H-1J and Figures 2C-E (now Figures 2E-G). Specific laser powers and energy densities are now consistently described in the figure legends, and we have added clarification in the main text to signal when we use each of the two light sources.

– It is not so clear whether it is indeed the change over time of the composition of the Optogranules after repetitive stimulation with blue light that is responsible for the induction of the cell death. Some additional experiments should be performed to clarify this issue. Is the dynamics of the formation and disassembly of Optogranules changing over time? It should also be investigated whether there is an effect of repetitive stimulation of the cells on the physicochemical properties of the stress granules. How is fluorescence for instance recovering after photobleaching?

Yes, importantly, we see evidence of maturation of granules such that their dynamics diminish with time. We believe that this may relate to the evolution of pathological inclusions. We have quantified this data and now present these results in new Figure 3—figure supplement 1C and 1D.

Regarding the reviewer’s question about “whether it is indeed the change over time of the composition of the OptoGranules after repetitive stimulation with blue light that is responsible for the induction of the cell death,” there was a clear correlation between repetitive induction of OptoGranules and cell death, but the mechanism of that cell death is unknown at this point. We believe that cell death is related to the OptoGranules because (to the extent possible) we have eliminated the confounding elements of exogenous stressors to induce granules, controlled for misexpression of fluorescently tagged proteins, and controlled for blue light exposure. Whether death relates to the *composition* of the granules is an interesting suggestion, but remains speculative.

– The added value of the iPSC related experiments is not so clear. It should be clarified in more detail in which sense the Optogranules are different in these cells in comparison to the Optogranules in the U2OS cells. At present, the overall impression is that what is concluded from these experiments is very similar (identical) to what was observed in the U2OS cells. If so, one wonders how the described process could lead to selective neuronal death.

We wanted to document the phenomenology of OptoGranule assembly and evolution in two different cell types, one of which has direct relevance to disease. Indeed, the results are qualitatively the same in U2OS cells and the iPSC-derived neurons, although within the “chronic persistent” paradigm, the neurons (Figure 4C) showed cell death much sooner than the U2OS cells (Figure 3C). With respect to cell type-specific vulnerability, we do not currently have sufficient evidence to conclude that neurons have increased vulnerability to repetitive stimulation of stress granules. It should be noted that aberrant stress granules are also linked to non-neuronal cell pathology, such as is observed in muscle diseases.

Minor comments:– The quality of some of the figures seems suboptimal (could be due to the conversion to a pdf). One clear example is the first figure illustrating the formation of Optogranules. While it is clear in the movie, it is much less clear to observe the individual droplets in Figure 1B.

We have optimized the images by reverting to the original high-resolution image files, including those shown in Figure 1B.

– It is indicated that the Optocontrol construct also forms a modest amount of nuclear and cytoplasmic clusters. It is not clear what the identity of these clusters is. Can this be determined and discussed in more detail?

We used two different Opto-Control constructs in this paper; one (“Opto- Control”) is a fusion protein of mCherry and wild-type CRY2, and the other (“Opto-Control (olig)”) is a fusion protein of mCherry and a mutant form of CRY2 (CRY2-olig) that forms abundant CRY2 clusters upon activation.

Like many proteins containing mCherry, the Opto-Control fusion protein undergoes a modest amount of nonspecific aggregation that is related to its degree of expression. Importantly, these clusters are negative for stress granule markers, including >10 proteins and polyA RNA (Figure 1H-J and data not shown). Furthermore, we affinity purified this protein and found by mass spectrometry that it does not associate with any stress granule proteins (data not shown).

To further challenge the negative results obtained with Opto-Control, we used Opto-Control (olig) to aggravate the aggregation of the fusion protein. The aggregates formed by Opto-Control (olig) were poorly dynamic by FRAP (new Figure 1—figure supplement 1D-F) and were negative for stress granule markers (new Figure 1—figure supplement 1C).

– No Opto-control is provided for the experiment in Figure 1G.

We now show the control experiment in new Figure 1—figure supplement 1C.

– Figure 1—figure supplement 2: The length of time that cells were exposed to blue light is not stated. Was this the same 6 h induction used in Figure 1?

In the resubmission, these images have now been incorporated into Figure 1—figure supplement 4. As detailed in the figure legend, cells were imaged after different time periods of blue light exposure, as appropriate. Images capturing live-cell GFP-G3BP1 fluorescence (panels B, C, I) were captured immediately after a 5-msec pulse by a blue-light laser. Images of cells that were fixed and stained for stress granule markers were captured after 2 hours (panels F, G) and 5 hours (panels D, E, J) of blue-light LED exposure depending on the time required for each marker to accumulate in OptoGranules.

– Figure 2A and 2B, quantification would be more convincing. e.g.% of cells with stress granules plotted over time after induction. The experiment should also be controlled using the Opto-control construct.

As noted above, quantitative information was collected in the course of conducting the experiments presented. With regard to the reviewer’s specific comments, we call attention to new Figures 2B and 2D.

– Figure 2C and D: It is concluded that the Optogranule assembly is independent of the regulation by eIF2alpha. This important observation should be discussed in more detail.

This is an important point that we now discuss more explicitly in the manuscript. Stress granule assembly may be initiated in response to a wide variety of stimuli, including viral infection, oxidative stress, heat shock, and proteasome inhibition. Recent studies have provided substantial insight into the molecular basis of stress granule assembly. Often, stress granule assembly is initiated through an accumulation of uncoated RNA in the cytoplasm. This can occur when stress signaling culminates in phosphorylation of the translation factor eIF2α, which inhibits translation initiation. Subsequent disassembly of translating polysomes and concomitant accumulation of uncoated, non-translating mRNAs favor stress granule assembly.

Alternatively, stress granule assembly may be initiated by oligomerization of the scaffolding protein G3BP, which bypasses the need for eIF2α phosphorylation. In brief, we have learned that stress granule assembly is driven by phase transition of G3BP with RNAs that have certain features (long, single-stranded, and limited secondary structure) (Yang, Mathieu et al., in review). When the collective protein-protein, protein-RNA, and RNA-RNA interactions breach a critical threshold, they undergo phase separation and initiate stress granule assembly. There are two ways to reach this threshold. The first and most common way is RNA-driven, as described above, and is based on increasing the concentration of RNA via inhibition of translation via phosphorylation of eIF2α. The second way to reach the critical threshold for phase transition is protein-driven, wherein oligomerization of G3BP increases valency for RNA binding. This latter mechanism is exploited in this manuscript, where oligomerization is forced through a CRY2- mediated interaction. As anticipated, this mechanism of stress granule assembly does require polysome disassembly, but does not involve the upstream step of phosphorylation of eIF2α.

– Figure 2E: Not clear whether the same blot was reprobed or whether different blots are shown for phospho-eIF2alpha and eIF2alpha. Why is the eIF2alpha at the top (and not in the middle) of the part of the blot that is shown?

In this figure panel (now Figure 2G) the same membrane was reprobed sequentially using antibodies corresponding to different species. The full membrane blotting data with serial antibodies are now shown in new Figure 2—figure supplement 1. Because the actin signal appears so closely above the eIF2α signal, the final cropped blot shows the eIF2α bands at the top of the cropped image. Methodological details have now been added to the figure legend.

– All stress granule components that were studied colocalised with the Optogranules. One obvious candidate that is missing in this long list is FUS (Results section). Was FUS present in these Optogranules?

Yes, FUS is present in OptoGranules, as now shown in new Figure 1—figure supplement 2G.

– Figure 3A: What is the effect of continuous blue light on the U2OS cells?

As shown in new Figure 3B, there is background toxicity associated with chronic exposure to blue light, which is unchanged by expression of Opto-Control. There is a significant increase in toxicity when cells express Opto-G3BP1, which we interpret as stress granule-dependent toxicity. Because of background toxicity associated with chronic exposure to blue light, we designed the “intermittent activation” paradigm to minimize the total dosage of light exposure, as described in the manuscript text.

– Figure 3C and D: From the Kaplan-Meier plots one can estimate that approximately 10-15 cells are followed after induction of the Optogranule formation. How were these cells selected and how many were selected in the three different independent experiments.

For the experiments shown in Figures 3C and 3D, we used live imaging to monitor stress granule dynamics in real time. To accomplish this, we used a 445-nm imaging laser and 60x Plan Apo 1.40NA oil objective, which provides a limited field of view. Opto- G3BP1 expression levels were measured using Prairie View software and cells with equivalent levels of Opto-G3BP1 expression were activated by blue-light laser. All activated cells were included in the analysis. All details are included in the figure legends (Figure 3C: n = 26 for Opto-Control and n = 28 for Opto-G3BP1; Figure 3D: n = 7 for Opto-Control and n = 10 for Opto-G3BP1).

– Figure 3C and D: Is there a statistical difference between the cell death induced by the induction of chronic persistent granules or after induction of chronic intermittent granules?

We found no statistically significant difference in the amount of cell death caused by the induction of chronic persistent granules compared to that caused by induction of chronic intermittent granules. These results are now shown in new Figure 3—figure supplement 1A. However, it should be noted that in the chronic persistent stress granule paradigm, the cells are exposed to much more blue light. Indeed, this is why we developed the chronic intermittent paradigm to minimize blue light exposure.

– Figure 3: On the Y-axis of panel c and d, the numbers are missing. It is also not clear whether the blue boxes are based on a quantification or whether these boxes represent what is expected (should be specified in the legend).

As now clarified in the legend, the left panels are schematics that represent the design of the light induction paradigm (blue boxes) and an idealized graph of the consequent cellular response (red line). Quantitative data is presented in the right panels.

– Figure 4: It is unclear whether persistent induction of Optogranules causes increased phosphorylation of TDP-43 or simply recruitment of a population which is already present in the cell. The authors should quantify the total levels of phospho-TDP-43 before and after chronic induction of Optogranule assembly to determine whether levels are increased.

Either of these mechanisms could account for the accumulation of phospho- TDP-43 in stress granules and our experiment does not permit us to make a distinction between them. Regrettably, although the amount of phospho-TDP-43 in cells can be detected on a cell-to- cell basis by immunofluorescence, this is not detectable by immunoblot.

– More details should be provided on the nature/quality of the neurons that are differentiated from the iPSCs. Are these cortical neurons? How pure are these neuronal cultures? Are these neurons electrically active?

For our study, we adapted the protocol published in Zhang et al., 2013, which produces cortical neurons, a cell type relevant to ALS/FTD. In the original Zhang et al. paper, the authors demonstrate that human iPSCs can be converted into electrically active functional neurons with nearly 100% yield and purity in 2 weeks using this protocol. To confirm that we successfully replicated the protocol of Zhang et al., we performed digital PCR to quantitatively analyze expression of the pan-neuronal marker *MAP2*, the pluripotent stem cell marker *OCT4*, the excitatory cortical neuron markers *BRN2* and *FOXG1*, and the synapsin marker *SYN1*. The results confirmed the cortical neuron identity of our cells, as now shown in new Figure 4—figure supplement 1A and B.

Additional data files and statistical comments:It is not always clear whether the correct statistics is used. One example is Figure 3B. A Student t-test doesn't seem to be the correct way to evaluate whether these differences are statistically different. Moreover, as these are box plots, it seems unlikely that the variation is indicated as standard deviations (s.d.) as is indicated in the figure legend.

We have examined the use of statistical tests throughout the manuscript and confirmed that the appropriate tests were used. In Figure 3B (note, this graph has been replaced with updated results that include parental U2OS cells), we now show the results as analyzed by two-way ANOVA, and have corrected the figure legend to reflect the fact that the variation is indicated as min to max.

Reviewer #3:

Very interesting paper – extremely easy to read and parse. Important for field, enjoyed reading, lots of interesting insights.Some comments:* It is hard to assess the degree of recruitment of different factors without any quantification (i.e. just from pictures). For example, it is not clear to be that there is bona fide p62 recruitment in neurons (Figure 4H). If this were a normal review I would insist this must be quantified to some degree (e.g. pixel mutual information using an intensity threshold as a binary mask, given intensities are hard to interpret).

Quantitative information was collected in the course of conducting the presented experiments. This quantitative data, with accompanying statistical analysis, has now been added to the figures. With reference to the reviewer’s specific comments, we call attention to new Figure 3I and Figure 4I, which show quantification of stress granule markers over time. For concerns relating to colocalization of p62/SQSTM1, we have now added line scans as new Figure 4—figure supplement 1E to illustrate colocalized signals.

* There are three distinct modes of illumination: continuous (Figure 3A), chronic persistent (Figure 3C) and chronic intermittent (Figure 3D). It is difficult to determine which is used in which experiment. I would strongly suggest standardizing this and ensuring each experiment is clearly labelled regarding which regime is used. Without this interpreting the data (for the reader) becomes difficult.

Indeed, there are three distinct induction paradigms used in this study, each with its advantages and disadvantages. For each light-based experiment, the corresponding induction paradigm is now clearly stated in the figure legend.

* The OptoXXX language used is a little confusing. In the Shin et al. paper, OptoDroplets refers to the general system of a Cry2 domain fused to something:"Here, we introduce an optogenetic platform that uses light to activate IDR-mediated phase transitions in living cells. We use this 'optoDroplet' system to study condensed phases driven by the IDRs of various RNP body proteins"Here, OptoGranules refer to the physical cellular assemblies (to distinguish them from stress granules – which makes semantic sense) and OptoG3BP1 is an example of an OptoDroplet construct that contains G3BP1. The fact that the OptoDroplet work is not used to introduce the technology is odd. The results determined here are of significant biological relevance and interest, but the approach is fundamentally the OptoDroplet platform applied to a different protein. The fact that OptoDroplets are only introduced once the OptoGranule system has been described:'These results contrast with observations made with light-induced aggregation of so-called "OptoDroplets," which nicely…'gives the impression that this is a different approach system, which is not the case (Figure 1A from this paper and 1A from Shin et al. are essentially interchangeable), and comes across as a little disingenuous. This was a general sentiment I heard from various places when the original preprint went up. It is surprising to me, because all of the exciting results pertain to the biology, not to the method, so it's unclear why the approach is introduced without basically saying "Here we extended the OptoDroplet system to examine the role of G3BP1 in nucleating stress granules…" or something to this effect.

We thank the reviewer for this frank comment. We coined the term “OptoGranule” with the *intention* of telegraphing that this structure extends and builds upon the approaches and observations described in Shin et al. for the “OptoDroplet.” Scientific advancements nearly always build upon prior scientific achievements, and it is appropriate (and polite) to acknowledge these prior influences. Optical induction of protein aggregation in live cells is a strategy that has been extensively employed. From our perspective, this is not the exciting and fundamental advance presented in Shin et al. Rather, these authors extend and build upon this prior work as a means *to control and monitor intracellular phase transitions*. As such, the OptoDroplet system is a valuable tool to study the biophysical phenomenon of phase transition and to delineate the roles of specific domains (particularly intrinsically disordered regions, or IDRs) in the formation of condensed phases.

The OptoGranule system further extends and builds upon this work by adapting the OptoDroplet approach to exploit recent insights into (1) the mechanism of stress granule assembly and (2) the distinct roles of *clients* and *scaffolds* in initiating assembly and maintaining specific RNP granule identity. As described above, the development of OptoGranules was informed by our identification of G3BP as a *uniquely essential central scaffold protein* for stress granules, in contrast to TIA1, TDP-43, FUS, and many others, which are *client* proteins (Yang, Mathieu et al., in review). Please note, this manuscript has been made available for assessment by the editors and reviewers. In the present manuscript we show that enforced phase separation of *client* proteins (TIA1, TDP-43, and FUS) using CRY2 generates OptoDroplets, which are relatively homogenous intracellular condensates, thereby confirming the results of Shin et al., but we also demonstrate that these are not stress granules. This makes sense since *client* proteins often reside in multiple biomolecular condensates with distinct identities. In contrast, enforced phase separation of a *scaffold* protein initiates the cascade of events that seeds the assembly of a full- fledged, complex stress granule, which we term an OptoGranule. Thus, an important takeaway from this work is the critical role of specific scaffolding elements in establishing and maintaining the identity of specific biomolecular condensates.

We stress that the distinction between “droplets” and “granules” is more than semantic. In part this distinction derives from our observation that Opto-TDP-43, Opto-FUS, and Opto-TIA1 form structures that do not recruit stress granule constituents, whereas those formed by Opto-G3BP1 appear to reconstitute the complex composition of an RNP granule. The second distinction is more subtle: droplets formed by Opto-TDP-43, Opto-FUS, and Opto-TIA1 have their biophysical origin in CRY2 oligomerization that presumably forces the IDRs of these proteins to self-associate and initiate a phase transition. In contrast, activation of Opto-G3BP1 forms granules because CRY2-based multimerization (specifically via the replaced NTF2L domain) increases the valency of G3BP, permitting it to engage with another scaffolding element (i.e., a class of RNAs), and these interactions create a seed that subsequently undergoes a phase transition that mediates subsequent further assembly of a stress granule.

Minor Comments:* It would be useful to draw the schematic (Figure 1A) with the domains proportional to their number of amino acids* Cry2+mCherry = ~840 amino acids (96 kDa), while NTF2 = 142 amino acids (16 kDa)* This, of course, may be impractical, in which case simply including the regional size in number of amino acids would be sufficient

To the extent practical given the limitations of the figure, we have addressed this suggestion by adding amino acid numbering to the schematic in Figure 1A.

* Is the absence of SGs in response to the double G3BP1/2 knockout a true absence of stress granules, or an absence of assemblies with G3BP1/Tia1/PAB1 markers – have other markers been checked?

This is an important point that we have fully explored (Yang, Mathieu et al., in review). We performed a genome-wide screen that identified G3BP as a uniquely essential factor in stress granule assembly. In follow-up studies, we created 23 individual cell lines in which putatively essential factors for stress granule assembly (among them G3BP) were knocked out using CRISPR-Cas9. Surprisingly, among these 23 cell lines, we found that *G3BP1/2* double knockout cells were the *only* cell line that failed to form stress granules in response to arsenite. To confirm this result, we stained *G3BP1/2* dKO cells for 21 different stress granule markers (G3BP1, G3BP2, caprin 1, PRRC2C, TIA1, USP10, ATXN2, CSDE1, eIF3η, PABP, TAF15, TRIM25, YB1, YTHDF1, YTHDF2, YTHDF3, TIAR, TDP-43, eIF4G, ataxin 2, and DDX3X, as well as polyA RNA), none of which accumulated into granules after arsenite treatment. We have now added data for TIAR and TDP-43 in new Figure 1—figure supplement 3A, bottom right panel.

* "…insinuate into stress granules and other membrane-less organelles" – not sure insinuate has an appropriate meaning here? 'Infiltrate' perhaps?

Although “insinuate” and “infiltrate” are often used interchangeably, we prefer “insinuate” here to evoke an underhanded and sinister infiltration that corrupts the organelle from within.

* Figure 1—figure supplement 1B is the same as part of Figure 1A?

Figure 1—figure supplement 1B was not the same as part of Figure 1A: it showed the design of the Opto-G3BP2 (rather than the Opto-G3BP1) construct. The point is now moot as we have removed Figure 1—figure supplement 1B.

* "Opto-Control expression remained diffuse, with a modest amount of nuclear and cytoplasmic clusters (Figure 1B and Supplementary Video 1)." – the text is intriguing, though it wasn't obviously clear to me where these cytoplasmic clusters are in Figure 1B? But assuming you do see some level of clusters, why is an obligate dimer able to lead to any degree of cluster formation? I would have expected dimers to be far far below the resolution limit such that monomer/dimer Opto-Control should be equally diffuse

As described in the response to Reviewer 1 above, we used two different Opto-Control constructs in this paper; one (“Opto-Control”) is a fusion protein of mCherry and wild-type CRY2, and the other (“Opto-Control (olig)”) is a fusion protein of mCherry and a mutant form of CRY2 (CRY2-olig) that forms abundant CRY2 clusters upon activation.

Like many proteins containing mCherry, the Opto-Control fusion protein undergoes a modest amount of nonspecific aggregation that is related to its degree of expression. Importantly, these clusters are negative for stress granule markers, including >10 proteins and polyA RNA (Figure 1H-J and data not shown). Furthermore, we affinity purified this protein and found by mass spectrometry that it does not associate with any stress granule proteins (data not shown).

To further challenge the negative results obtained with Opto-Control, we used Opto-Control (olig) to aggravate the aggregation of the fusion protein. The aggregates formed by Opto-Control (olig) were negative for stress granule markers (new Figure 1—figure supplement 1C) and poorly dynamic by FRAP (new Figure 1—figure supplement 1D-F).

* When the OptoControl is off it appears to partition into the nucleus (Figure 1H, 1I, 1J) but when activated it is strongly excluded. Both monomeric and dimeric version should be larger than the threshold size for free permeability into the nucleus sans an NLS, so it’s not clear how/why light should be changing the distribution – some explanation of this would be useful (i.e. does this imply there are additional un-characterized responses to blue-light stimulation?).

We don’t know, but this effect could be related to a report (Pathak et al., Nucleic Acids Res, 2017; PMID 28431041) that CRY2-mCherry shows minor protein redistribution from the nucleus to the cytoplasm following blue light stimulation. It is possible that light-induced CRY2 cluster formation may block or slow nuclear import of cytosolic and newly synthesized protein, leading to accumulation of protein in the cytosol.

* Are the expression levels of OptoG3BP1 comparable to endogenous G3BP1? I realize expression is clearly a tunable parameter as addressed in Figure 2, so it would be useful, as a reference, to mention the expression levels of endogenous protein here.

We selected Opto-G3BP1 expression levels that were comparable to endogenous G3PB1, as shown in new Figure 1—figure supplement 1A.

* "Expression of Opto-constructs using full-length TDP-43 or FUS [CRY2PHR-mCherry-TDP-43(FL); CRY2PHR-mCherry-TDP-43(FL)] did not produce cytoplasmic clusters with blue light activation" (minor point – I believe the second instance here should be CRY2PHR-mCherry-FUS(FL))

We have now corrected this.

* This is a very interesting result for two reasons* 1) It’s not clear to me why FL-FUS OptoDroplets should not form assemblies, especially given recent reports that FL FUS assemblies much more readily than the LCD alone. Any thoughts on this in the discussion would, I think, be very useful. [1]

This is a major point of this paper. The difference relates to the distinct roles of *clients* and *scaffolds* in initiating assembly and maintaining specific RNP granule identity. This point is explored above and now made more plain in the manuscript text.

* 2) Recent work has shown that the NTD of TDP43 forms dimers that are necessary for phase separation in full-length TDP-43. In agreement with this, the CRY2 dimerization domain appears able to replace this N-terminal domain and give rise to droplets, HOWEVER, if the N-terminal domain is present (i.e. in CRY2PHR-mCherry-TDP-43(FL)) droplet formation is suppressed – any possible mechanism here would be useful to address – specifically, how does the ADDITION of a dimerization domain between CRY2 and the C-terminal portion of TPD43 suppress assembly?* These questions are directly related to the comparison of other optoDroplet constructs

To be clear, we do not claim that CRY2_PHR_-mCherry-TDP-43(FL) does not form droplets; rather, we claim that this fusion protein does not form stress granules. Indeed, this is the very distinction between OptoDroplets (phase separation of a selected protein) versus OptoGranules (a cascade of events, including polysome disassembly, that leads to assembly of a stress granule). As the reviewer notes, we did observe that CRY2_PHR_-mCherry-TDP-43(FL) forms puncta in the nucleus, which corresponds to the normal localization of TDP-43 and is consistent with the presence of an NLS in the NTD (new Figure 1—figure supplement 4G).

Upon activation, these puncta are indeed less prominent for unclear reasons. We speculate that the presence of two dimerization domains (CRY2 and NTD) leads to tighter intermolecular interaction between any two TDP-43 molecules and this tight interaction makes TDP-43 unfavorable for weak multivalent interactions that are favorable for LLPS.

* Not sure how to interpret the Opto-G3BP1 vs G3BP1-GFP RNA binding data? Naively from the description of the RNA binding assay I would have expected 1) cross-linking to cause G3BP1 in stress granules to extensively cross-link with local RNAs, while under non-stress/non OptoGranule inducing conditions there would be minimal RNA cross-linking and 2) for Opto-G3BP1 and G3BP1-GFP to be similar. Instead, the data appear to show that you get extensive and essentially identical RNA binding under stressed and non-stressed conditions, and that Opto-G3BP1 and G3BP1-GFP have different RNA binding profiles. Similarly, not sure how to interpret the western blot associated with 2b – it would be useful to explicitly annotate what the bands are. This is almost certainly just a limitation of my understanding of the experimental setup, but it seems at least plausible others might also be confused…

The purpose of this experiment was to show that Opto-G3BP1 binds RNA to a similar extent as WT G3BP1. Indeed, the total amount of RNA bound by G3BP before and after stress is unchanged when assessed by CLIP (cross-linking immunoprecipitation). What is not revealed by this experiment, however, is that the types of RNA species bound by G3BP1 are shifted by stress, and this shift appears to be important in initiating stress granule assembly.

These results are elaborated in Yang, Mathieu et al. (in review) and are out of context here, so we have deleted these results from this manuscript.

* In the protein name SQSTM1 is used, while in Figure 3 p62 is used – this may confusing for people unfamiliar with the protein, so I'd suggest a consistent naming scheme

We now use SQSTM1 uniformly throughout the manuscript.

* I found the mCherry vs. mRuby optoG3BP1 section confusing on the first read, as it was initially unclear why both would be needed. I would suggest splitting this into a 'first we used mRuby-optoG3BP1…', then once this has been completed, explain why an inducible system is needed and then introduce the inducible mCherry variant.

We have rearranged this portion of the text to increase clarity.

* Why is the pTDP-43 recruitment to SGs in neurons so much weaker than in U2OS cells? In U2OS see reasonable recruitment just 1 hour in, with robust recruitment after 2 hours. In neurons after 5 hours (4g) there is basically no co-localization with optoG3BP1, and then after 6 robust co-localization is observed.

We do not know why these differences occur. We can only point to inherent differences that must be present in the cellular properties of cortical neurons versus those of the U2OS cell line.